# Seasonal regimes of warm Circumpolar Deep Water intrusion toward Antarctic ice shelves

Joshua Lanham [1] ✉, Matthew Mazloff[2], Alberto C. Naveira Garabato[3], Martin Siegert [4] & Ali Mashayek[1]

Basal melting of Antarctic ice shelves is primarily driven by heat delivery from warm Circumpolar Deep Water. Here we classify near-shelf water masses in an eddy-resolving numerical model of the Southern Ocean to develop a unified view of warm water intrusion onto the Antarctic continental shelf. We identify four regimes on seasonal timescales. In regime 1 (East Antarctica), heat intrusions are driven by easterly winds via Ekman dynamics. In regime 2 (West Antarctica), intrusion is primarily determined by the strength of a shelf-break undercurrent. In regime 3, the warm water cycle on the shelf is in antiphase with dense shelf water production (Adélie Coast). Finally, in regime 4 (Weddell and Ross seas), shelf-ward warm water inflow occurs along the western edge of canyons during periods of dense shelf water outflow. Our results advocate for a reformulation of the traditional annual-mean regime classification of the Antarctic continental shelf.

The future of floating Antarctic ice shelves is one of the principal uncertainties surrounding projections of future sea level rise[1]. A major control on the response of Antarctic ice shelves to anthropogenic climate change is the dynamics governing the water mass structure on and near the continental shelf. The majority of basal melting is caused by relatively warm (conservative temperature ($\theta$) $\geq 0\,°C$) and salty (salinity ($S$) > 34.5) Circumpolar Deep Water (CDW)[2]. CDW is supplied to the near-Antarctic Southern Ocean by the large-scale circulation[3,4]. It upwells from depth along outcropping isopycnals, mixing with cooler waters on the continental shelf to ultimately cause melting[5]. However, the exact set of processes by which this happens—and the routes along which CDW intrusion onto the shelf occurs—remain little understood.

The formation of Dense Shelf Water (DSW) at the ocean-sea ice interface is known to be a key factor in controlling the amount of warm CDW that can cross the shelf break[6]. DSW is a high-salinity ($S$ > 34.5), cold ($-1.9\,°C < \theta < -1.8\,°C$) water mass formed at the Antarctic margins from the salt rejection associated with sea ice formation[7]. DSW plays a key role in abyssal ventilation and the overturning circulation, as it is implicated in the production of Antarctic Bottom Water (AABW) in overflows at the continental shelf break[8]. Importantly, the presence of DSW on the continental shelf limits the intrusion of CDW onto the continental shelf[9,10]. Production of the high-density DSW is induced by strong vertical convection and an almost complete destratification during the winter months. This floods the continental shelf with DSW, forming a meridional transport barrier to the lower-density CDW. DSW production sites are highly localised, with most production occurring in the coastal polynyas of the Weddell and Ross seas, the Adélie Coast and Prydz Bay[11–13].

Wind stress also modulates CDW access to ice shelves. Low-level Antarctic easterly winds induce shoreward Ekman transport and downwelling of cold and fresh surface waters, producing a sharp isopycnal front known as the Antarctic slope front (ASF) above the continental slope. Observational data show that the strength of these easterly winds can influence the supply of warm waters to the near-ice shelf region: a weakening of the easterly winds reduces coastal downwelling, flattening isopycnals and enabling CDW to access the continental shelf[14]. The Antarctic slope current (ASC) provides another dynamical barrier to CDW intrusion. It flows westward along the ASF in response to the sea surface height (SSH) gradient produced by easterly wind stress[15]. The sloping isopycnals of the ASF counter the barotropic SSH gradient, slowing the flow at depth in accordance with thermal wind balance. The ASC exhibits substantial spatio-temporal variability[16] and becomes bottom-intensified in regions of DSW formation where the direction of isopycnal slope reverses[17]. Various studies have highlighted the importance of transient flows, such as bottom-trapped eddies and tides, in enabling CDW to cross the ASF/ASC and reach the continental shelf[18–20].

Bottom topography is also important in modulating CDW intrusion onto the Antarctic continental shelf. Steering of deep water toward and into the ice-shelf cavity in troughs is known to control melt rates[21–23], whilst the configuration of the grounding line determines the exposure of the ice shelf to CDW and its susceptibility to marine ice sheet instability[24]. The geometry

[1]Department of Earth Sciences, University of Cambridge, Cambridge, UK. [2]Scripps Institution of Oceanography, University of California, La Jolla, USA. [3]National Oceanography Centre, University of Southampton, Southampton, UK. [4]Penryn Campus, University of Exeter, Penryn, UK. ✉e-mail: jal238@cam.ac.uk

of the ice shelf and its cavity are also important in determining the strength of blocking of poleward barotropic CDW inflow[25]. Amblas and Dowdeswell[26] show that the continental shelf depth, width and trough anatomy shape DSW outflow strength and location.

The localisation of DSW production, spatial variability in mean easterly wind strength, and heterogeneity of bottom topography all culminate in contrasting regimes of water mass structure around the Antarctic margin. Regions with strong DSW formation, such as the Weddell and Ross seas, are examples of 'cold' shelf regimes[12,18,26,27]. These are characterised by a strong V-shaped ASF and a low level of CDW intrusion onto the continental shelf[9,18,28]. An example of this regime in the Weddell Sea is visualised in 3D in Fig. 1. In contrast, 'warm' regimes with little or no DSW production and weak easterly winds have a tenuous or non-existent ASF, enabling CDW to flow directly onto the continental shelf. This regime is typical of the Amundsen and Bellingshausen seas in West Antarctica, which experience relatively weak coastal easterly winds[12,18,27,29]. Amundsen Sea temperatures are visualised in Fig. 1. West Antarctica therefore exhibits substantially higher melt rates than East Antarctica[30]. Warm shelf regions are often considered to be more sensitive to future anthropogenic forcing in climate projections[31,32], due to both a projected weakening in west Antarctic easterlies[33], and changes in buoyancy forcing[34,35]. Intermediate or 'fresh' regimes occur when strong easterlies induce Ekman transport and downwelling, but there is little-to-no DSW production. In these cases, Low Salinity Shelf Water (LSSW) or Ice Shelf Water (ISW) is the dominant water mass near the ice shelf. Intermediate regimes occur throughout East Antarctica, for example in the Knox and Princess Martha coasts[12,18].

The response of shelf regimes to climate change is likely to partially hinge upon changes in the atmospheric circulation around Antarctica. The Southern Annular Mode (SAM)—measuring the intensity and meridional position of mid-latitude westerlies—is projected to tend towards an increasingly positive phase, due to an anthropogenic diminishing of the equator-to-pole thermal gradient[33,36,37]. Positive SAM years bring westerlies further south, weakening the easterlies around Antarctica, reducing ASF strength, and thereby increasing CDW flux onto the shelf. An increasingly positive SAM over recent decades is well documented in the observational record, and is attributable to greenhouse gas and ozone forcings[38,39]. Despite this, a circumpolar-mean weakening in Antarctic easterlies is not evident in observations. Instead, reanalysis data show an intensification of the seasonality of winds: easterlies have weakened in summer and strengthened in winter[39]. Given that summer is the main season for basal melting across most of the Antarctic margins[40–42], a weakening of the easterlies and a corresponding warming on the shelf during this season is likely to have considerable impacts for future climate.

Model simulations also indicate that meltwater forcing will play a leading-order role in future change along the Antarctic margin. Meltwater flux from melting ice suppresses the formation of DSW and the associated AABW production and abyssal ventilation[6]. Moorman et al.[27] find that this can induce both positive and negative feedback effects on the direction of melting. A reduction in the salinity of shelf waters in DSW-producing regions causes an increase in stratification, which reduces vertical heat transport and leads to sub-surface warming. However, input of freshwater near the shelf steepens the density gradients associated with the ASF, thermally isolating the shelf from warm offshore CDW. Contrasting representations of these opposing processes leads to disagreement in models, which has been attributed to the representation of the ASC[43]. At a larger scale,[44] show that future meltwater input near the shelf results in a contraction in the AABW layer, facilitating a poleward expansion of CDW that opens new pathways for warm intrusions onto the continental shelf.

To gain physical understanding of the full array of CDW intrusion-governing processes, we require a resolution that is not available in

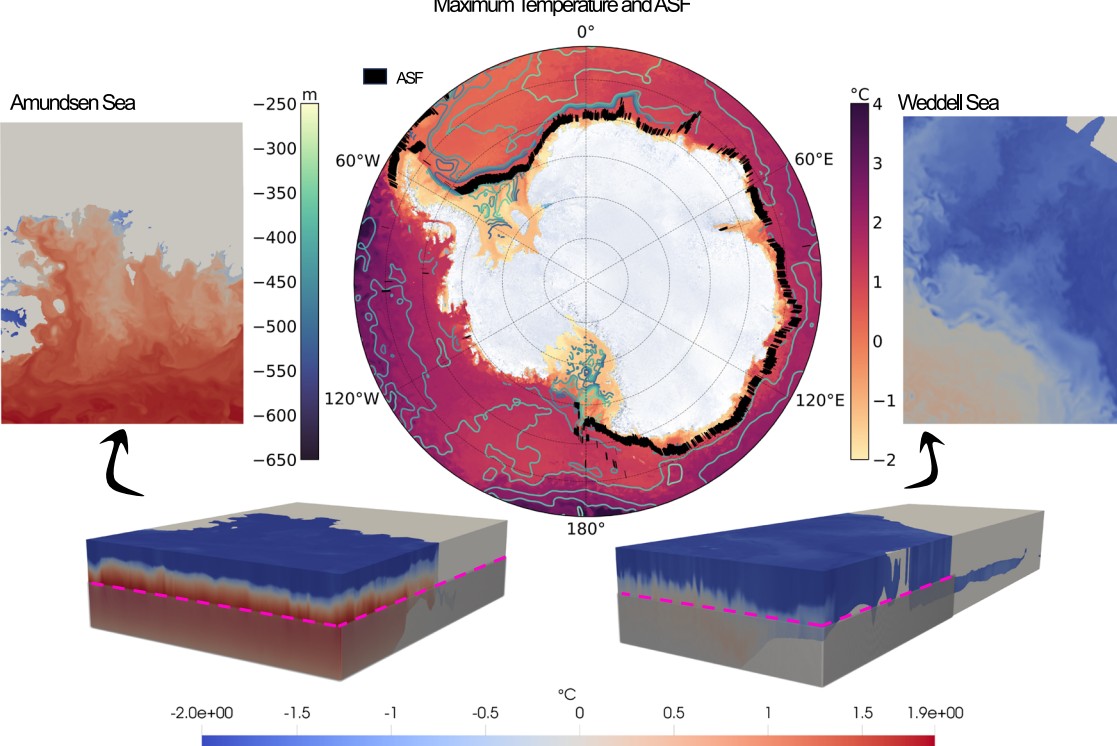

**Fig. 1 | The Southern Ocean high resolution model (SOHI).** Pictured centrally is the maximum temperature in the water column (yellow-red colormap) and the Gaussian-filtered depth at which this maximum occurs (blue-green contouring). Both the temperature maximum and depth are consistent with the properties of Circumpolar Deep Water. A metric for ASF detection is shown in black (for more information on the method see Supplementary Information, Section S1.1 and Fig. S2). The 3D visualisations illustrate the near shelf regions of both the Amundsen (left) and Weddell (right) seas. Shelf cavities are visible in both plots. 2D slices of the temperature field from the dotted pink line illustrate the horizontal resolution of the model and the contrasting regimes at each shelf. All plots use a 24-h average on September 1st, 2005.

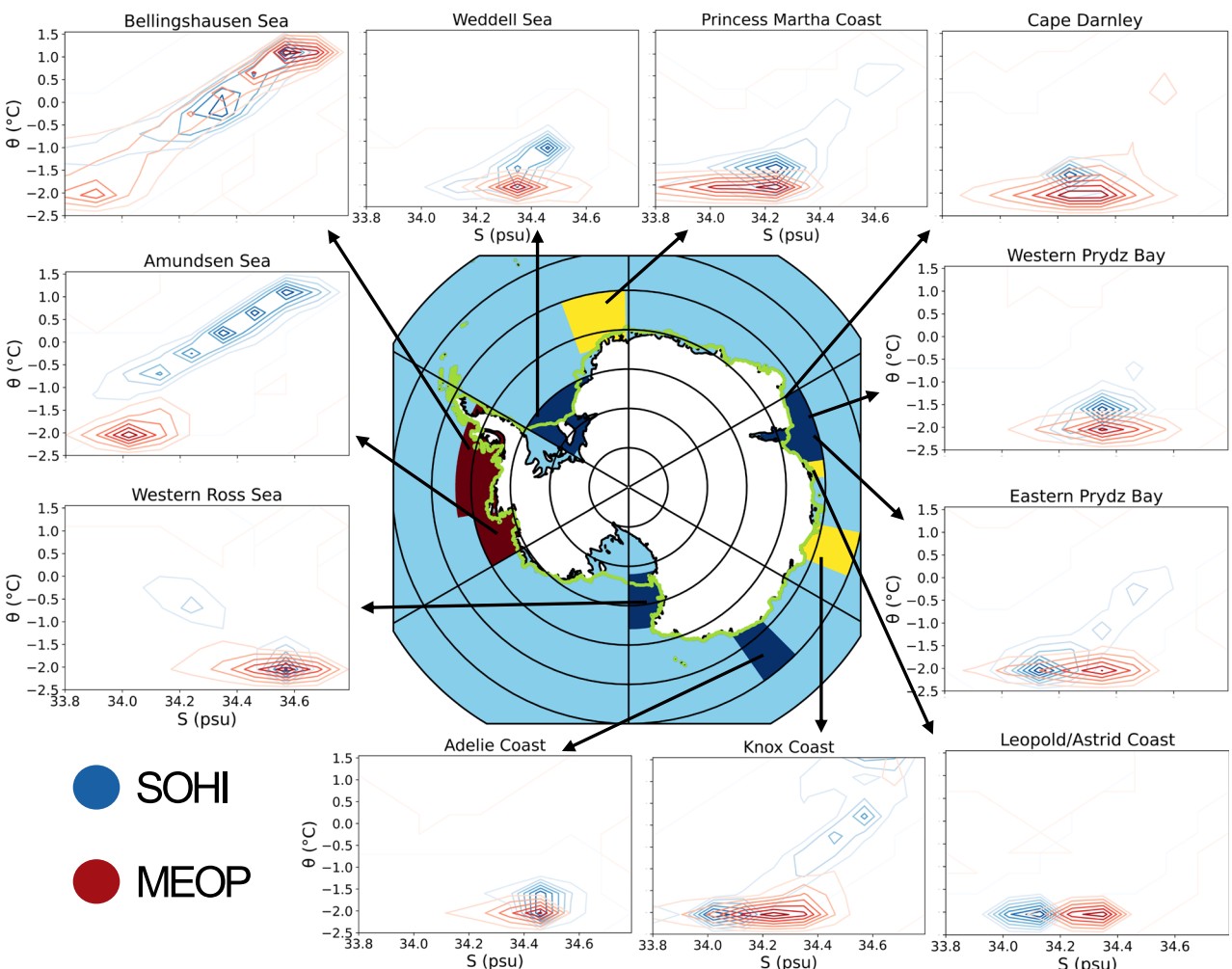

**Fig. 2 | Verification of SOHI's near-shelf regions with MEOP seal data.** Contour plots show bivariate temperature-salinity histograms of both SOHI (blue) and MEOP (red). MEOP equivalent measurements are sub-sampled from SOHI as the nearest grid point at that date and time. Whilst this removes the seasonal signal, data scarcity constraints mean that all MEOP years are included, such that the sampling method may unintentionally include the impact of interannual variability. These plots are shown for a range of near-shelf regions that are defined by Narayanan et al.[12]. The regions are selected to reflect the sampling density of MEOP data. The time-mean regime classification, according to Narayanan et al.[12], is shown by each region's colouring (dark red indicates a warm regime, yellow a fresh, and blue a cold). The northern limit of the ice shelf is delineated in green.

operational climate models. We thus use the Southern Ocean high-resolution (SOHI) model (Fig. 1), which is constrained by observational data assimilated by its parent model, the Southern Ocean State Estimate (SOSE). SOSE is a massive data assimilation effort with great resource costs, but it cannot fully resolve the suite of processes necessary to understand the complexity of seasonal CDW intrusion. SOHI is initialised from SOSE in order to better resolve these dynamics. It has hourly wind forcing and tides and, at 1/24th degree horizontal resolution with 225 vertical levels, is able to capture many of the mesoscale/sub-mesoscale dynamics that are known to be important for cross-ASF transport of CDW. The resolution also enables a high-complexity representation of the continental shelf bathymetry and ice shelf cavity. Hydrographic near-shelf verification of SOHI is provided in Fig. 2 and is further discussed in Section "Verification against observations".

Understanding the seasonal patterns of CDW intrusion onto the Antarctic shelf is integral to producing robust projections of Antarctic ice sheet stability. Shelf regimes are often conceptualised in terms of their mean state conditions (i.e. cold, warm or fresh). Here, we build upon these classifications to define the regimes of seasonality of CDW intrusion that characterise the Antarctic margins, and shed some light on which regions are likely to be more vulnerable to future perturbations in seasonal atmospheric forcing. We provide a unified view of CDW intrusion along the Antarctic margin, integrating results from a variety of different regional studies to shed light on why and how these vary from region to region.

## Results

We classify water masses in SOHI with the method outlined in Section "Model and methods". The maximum fractions of CDW and DSW at the bottom of each model location across the seasonal cycle are shown in Fig. 3a and c, respectively (see Fig. S1 for a full sample output). The analysis successfully reproduces the four principal known areas of DSW formation: the Weddell and Ross seas, the Adélie Coast and Prydz Bay. CDW concentrations are consistent with our understanding of Antarctic shelf regimes; higher concentrations are present across the West Antarctic warm shelves, whilst a sharp front is visible near the continent across the cold and fresh regimes of East Antarctica. The ASF, displayed in Fig. 1, is also clearly identifiable in the depth-latitude cross sections shown in Supplementary Information (see Fig. S1).

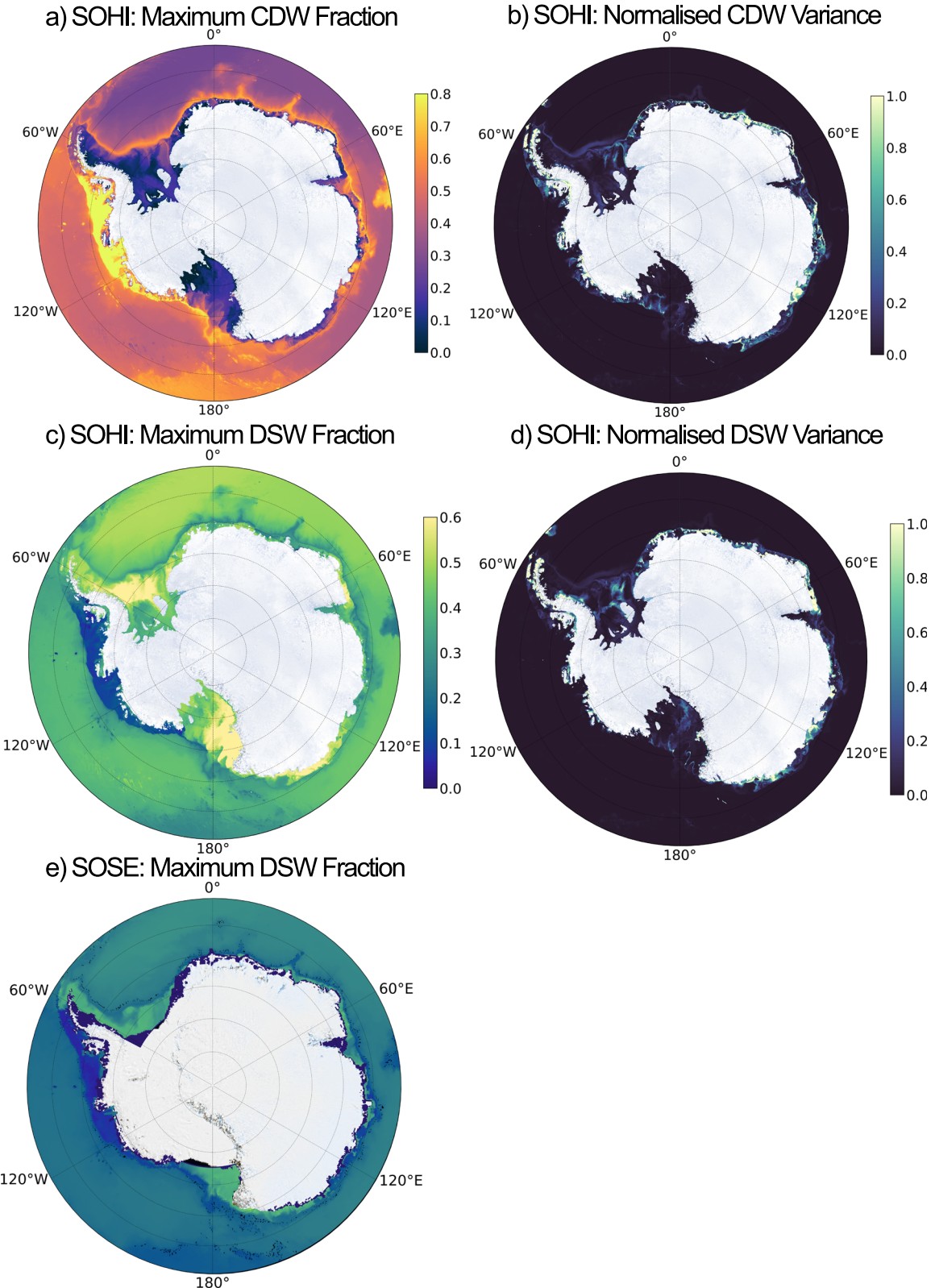

**Fig. 3 | CDW and DSW around the Antarctic margins in SOHI and SOSE.** Panel **a** shows the maximum fraction of CDW at each bottom grid point across the annual cycle. Warm regimes, such as West Antarctica, are identifiable as shelf regions with high maximum CDW fractions. Cold regimes, such as the Ross and Weddell seas, exhibit a near-zero maximum fraction of CDW. Panel **b** shows the normalised variance of the annual time-series of CDW fraction at the bottom grid cell. Variance of the CDW fraction time-series is computed and normalised by the 99th percentile.

High values indicate regions in which there is a high degree of seasonality of CDW on the shelf. Panels **c**, **d** are the same as **a** and **b**, but for DSW. Panel **e** is the same as **c**, but uses output from 1/6° SOSE, with identical Source Water Types (see Section "Water mass classification: machinery and source water type selection"). SOHI simulates the known areas of DSW formation well, whereas DSW production in the lower-resolution SOSE is markedly reduced.

**Fig. 4 | Timing of on-shelf CDW maxima and correlation with zonal wind velocity.** Panel **a** shows the month in which the local bottom CDW fraction maximum (Fig. 3a) occurs, and Panel **b** shows the correlation between the bottom time-series of monthly-mean CDW fraction and the monthly-mean time-series of 10-m ERA-5 zonal wind velocity from the nearest surface grid point. Both panels are masked by the seasonal variance from Fig. 3b, only showing shelves in which the normalised variance exceeds 0.27. Stippling shows regions in which the *r*-value is significant at the 90% level, assuming 10 degrees of freedom. *r*-Values are significant at the 99% level in some parts of the shelf between 30–60°E, in the western Amundsen Sea and also in Vincennes Bay to the west of the Browning Peninsula. These areas are also significant at the 90% level when reducing the degrees of freedom to between 4 and 8 (see Fig. S4). At the 95% level, the significance in this area expands to include the Totten Glacier (100–120°E), which is also significant at the 90% level assuming 8 degrees of freedom (see Fig. S4).

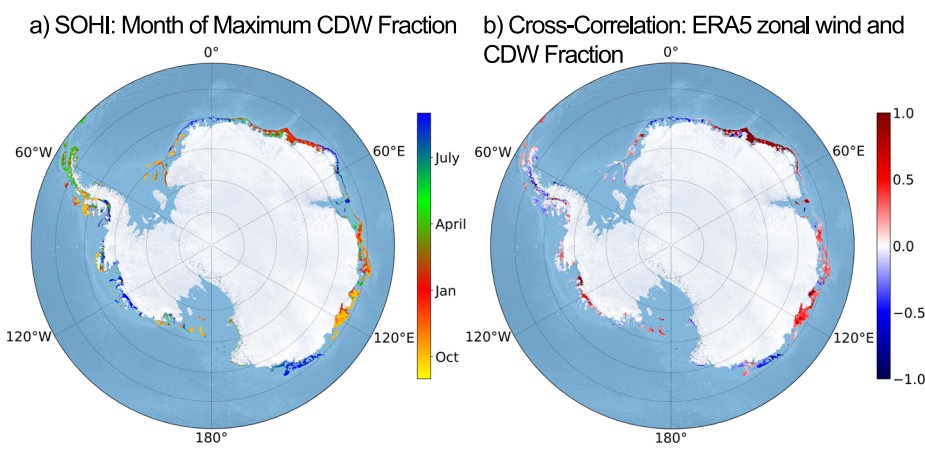

a) SOHI: Month of Maximum CDW Fraction

b) Cross-Correlation: ERA5 zonal wind and CDW Fraction

The analysis is repeated for DSW in SOSE. This is shown in Fig. 3e, which reveals that SOSE does not simulate the necessary θ–S range required to produce the realistic pattern of DSW formation present in SOHI. The lack of cavities means that SOSE misrepresents the Antarctic coastal processes implicated in DSW production in the Ross and Weddell seas. This is consistent with a wide range of studies that show that coarse-resolution models—such as coupled climate models—are incapable of a realistic representation of DSW or AABW formation[45–48].

To identify the shelf regions that experience considerable levels of seasonal CDW intrusion, we compute the variance in CDW concentration at the sea floor. This is normalised by the 99th variance percentile and shown in Fig. 3b. We choose the sea floor as a depth level in order to make calculations computationally feasible, whilst capturing all horizontal locations in the model (including within the cavity). This assumes that all CDW intrusions onto the shelf have at least a component at the sea floor. The water mass classification outputs in all sample regions studied shows that this is a good assumption (see Fig. S1), however, we acknowledge that this method may miss some CDW intrusions that occur above the sea floor (particularly in regions of DSW overflow).

The month at which the local maximum in CDW at the bottom occurs is shown in Fig. 4a, filtered by regions of high seasonality according to the variance in Fig. 3b. Summer CDW intrusions (shown in red) occur along the shelf throughout the majority of East Antarctica. There are notable exceptions including the far eastern Weddell Sea and the Adélie Coast, which show winter maxima (blue). The shelves of the Bellingshausen and Amundsen seas are also generally characterised by winter maxima in CDW occurrence. That being said, we note the presence of trends in CDW fraction at the Adélie Coast, and trends in both CDW fraction and wind velocity in the West Antarctic shelf and some parts of East Antarctica (principally 80–120 °E), such that we suggest the timing of CDW maxima in these regions to be treated with some caution (see Section S6).

We use a correlation analysis to help with the classification of regimes, with the typical caveat that correlation does not necessarily imply causation. Figure 4b displays the correlation between the monthly-mean ERA5 zonal wind velocity and the CDW concentration at the seabed. Throughout East Antarctica, eastward winds are positively correlated with CDW concentrations on the continental shelf. The West Antarctic shelf also exhibits positive correlations through the Bellingshausen and Amundsen seas. Notable negative correlations occur in the far eastern Weddell Sea and Adélie Coast.

Abyssal depths are naturally identified in Fig. 3b as regions of near-zero seasonal variability away from the continental shelf. However, there are also shelf regions that appear to show little-to-no seasonality in bottom CDW concentration. These are primarily the non-canyon areas of the Ross and Weddell seas and Prydz Bay. We suggest that this is largely the result of shelf bathymetry. Recent work has shown that the large shelf extent of the Ross and Weddell seas enables DSW build-up over years-to-decades[13], resulting in a large reservoir of DSW that prevents or minimises the seasonality of DSW export and CDW inflow. In contrast, narrower shelves, such as Prydz Bay, show high levels of seasonal export of DSW, illustrated by the DSW seasonal variance plot in Fig. 3d. We attribute the lack of seasonality in Prydz Bay to the particularly shallow depth of the shelf and this is discussed in more detail in Section "Buoyancy-driven regimes". There are also shelf regions in West Antarctica that exhibit little-to-no seasonality, most notably the Eltanin Basin in the Bellingshausen Sea. In this case, we suggest that a lack of seasonality stems from the absence of sustained seasonal wind forcing. Easterly winds are not sufficiently intense to sustain an ASF/ASC, such that isopycnals at the shelf break remain flat, nor are westerly wind anomalies sufficiently consistent to induce an undercurrent as in the Amundsen Sea (Section "Wind-driven regimes").

Of the continental shelf areas that exhibit seasonality in CDW concentrations, we classify regions into wind-driven and buoyancy-driven regimes of seasonal CDW intrusion. A schematic summarising these regimes is shown in Fig. 5. Wind-driven regimes are either governed by ASF intensification/relaxation shown as the 'ASF Regime' in Fig. 5 (Section "Wind-driven regimes"), or by an acceleration/deceleration of the undercurrent shown as the 'Undercurrent Regime' in Fig. 5 (Section "Wind-driven regimes"). The ASF sub-regime dominates in East Antarctica, as shown in Fig. 5. CDW concentration maxima in summer occur when the mean along-slope easterlies weaken and the ASF relaxes, giving rise to a positive correlation between CDW fraction and ERA5 zonal wind velocity. Positive correlations are also dominant on the West Antarctic shelf, but we find that the primary agent of variability here is the along-slope undercurrent (Fig. 5).

Canyons in the Ross and Weddell seas, along with the entire Adélie Coast shelf, are characterised as buoyancy-driven. We attribute the seasonal variability in CDW intrusion onto the continental shelf in these regions as being primarily related to the formation and export of DSW near the shelf break. Buoyancy-driven regimes are sub-divided into (a) the direct 'blocking' influence of the DSW formation cycle shown as the 'DSW Export Regime' in Fig. 5, and (b) DSW outflow initiating CDW inflow along canyons via barotropic dynamics shown as the 'Canyon Regime' in Fig. 5. We find that the Adélie Coast falls into the first of these sub-regimes, and

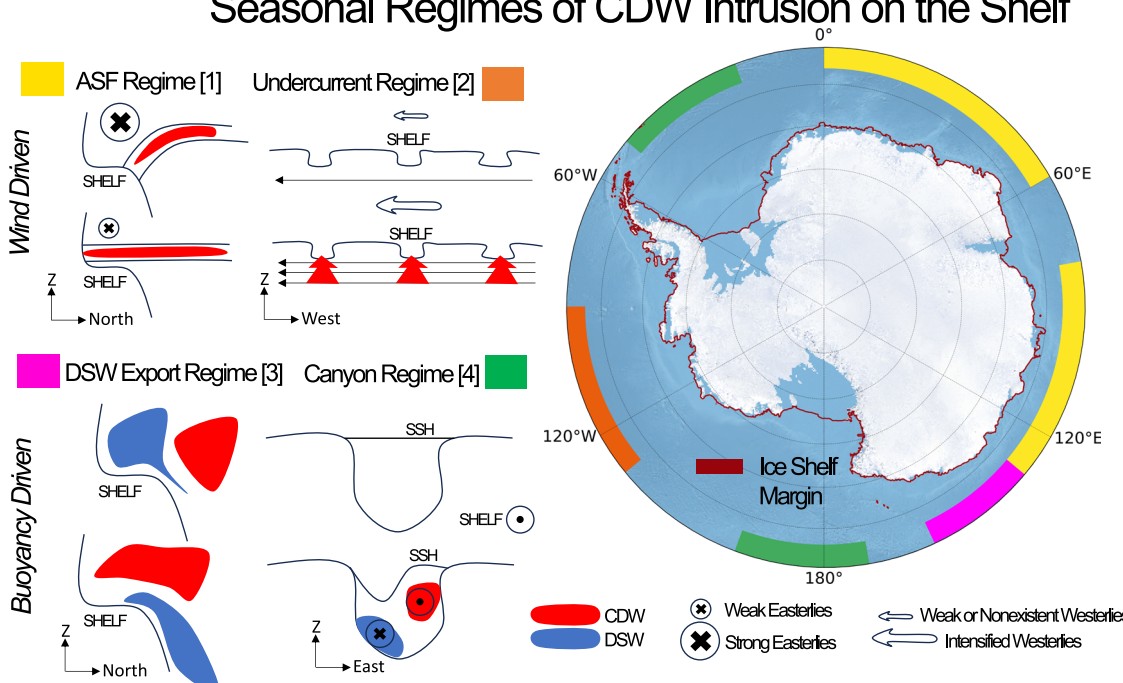

**Fig. 5 | Summary figure illustrating schematically each of the four regimes identified in this study, along with their locations.** The regimes are enumerated and related to the numbering of the regimes in the Abstract. Regimes 1 and 3 are depth-latitude cross sections, whilst 2 and 4 are depth-longitude cross sections. The ice shelf margin is shown in red. In Regime 1, isopycnals (black lines) flatten to let CDW access the shelf. This is driven by a weakening in the strength of along-slope easterly winds. In Regime 2, the undercurrent (horizontal eastward arrow) accelerates due to a period of (Amundsen low-induced) sustained westerly winds, causing uplifted isopycnals and increased transport of CDW on to the shelf via troughs. In Regime 3, CDW can access the continental shelf only when DSW has been exported. In Regime 4, the outflow of DSW along the western edge of the canyon induces a SSH gradient which promotes shelf-ward CDW inflow along the opposing side.

that this may account for the negative correlations over this shelf in Fig. 4b. This is discussed further in Section "Buoyancy-driven regimes".

The negative CDW-wind correlations in the far eastern Weddell Sea are suggested to be the result of CDW advection by high westward ASC velocities in this region[49] (see Fig. S3). This is discussed in the Supplementary Information in Section S2. We neglect to classify the Antarctic Peninsula, despite its hosting of substantial seasonal CDW variability. This is primarily because observational evidence in this region does not support a strong systematic seasonal cycle, instead suggesting that the majority of CDW reaches the continental shelf via (sub-)mesoscale eddies from the ACC[50–52]. Further justification of this choice can be found in Section S3 in the Supplementary Information.

### Wind-driven regimes

**ASF mechanism.** Antarctic easterly winds encircle the continent at low levels[29]. The strength of these winds is a major factor in determining the amount of CDW that can access the continental shelf, via the modification of the cross-slope density gradient and along-slope velocity of the ASF (Section "Introduction"). On average, Antarctic easterlies across East Antarctica weaken in austral summer, relaxing the ASF, reducing coastal downwelling of cold surface waters and expectedly enabling enhanced net on-shelf transfer of CDW[39,53]. Comparison between both the climatological and 2005/2006 summer-winter anomalies in ERA-5 zonal wind (see Fig. S5) illustrates that the seasonal wind forcing during 2005/2006 was largely typical of the climatological mean. A more detailed discussion can be found in Section S3 in the Supplementary Information.

*r*-Values in Fig. 6 exhibit a coherent positive signal throughout East Antarctica. This correlation is significant at the 99% level over 20–60°E (see Fig. S4), and at the 90% level over parts of 80–130°E (Knox Coast, Sabrina Coast) and the entrance to the Amery Ice Shelf. This suggests that CDW concentrations respond to the seasonal wind forcing cycle via changes in the isopycnal configuration. An illustrative cross-section is shown in Fig. 6. As easterly wind forcing reduces in summer, the ASF relaxes and enables increased net on-shelf transfer of CDW. In the region 20–60°E, where correlations are strongest, the seasonal cycle of CDW on the shelf closes (see Fig. S6a), such that we can be confident in the timing of the summer CDW maxima in this region. In the region 80–130°E, there is a negative trend in both the CDW and zonal wind time series (Fig. S6d and f), such that the timing of the CDW maxima in this region should be considered less robust. With the notable exceptions of Prydz Bay and the Adélie Coast (Section "Buoyancy-driven regimes"), we classify East Antarctic CDW seasonality as 'wind-driven' via an ASF-mediated response.

**Undercurrent mechanism.** The mechanism driving seasonal CDW transport toward the West Antarctic ice shelves is distinct from the ASF mechanism. We attribute the seasonal variance at the shelves of the western Bellingshausen and Amundsen seas (Fig. 3b) to the wind-driven strength of a shelf-break undercurrent in these regions, which is known to drive poleward CDW transport there[24,54,55].

The seasonal pattern of wind forcing is seemingly reversed in West Antarctica: the south-westward migration of the Amundsen Sea low-pressure system during austral winter causes westerly wind maxima in winter, with easterly wind maxima in summer (see Fig. S5)[56]. The implications of this are twofold: firstly, it is possible that coastal westerly winds may increase CDW fractions on the shelf through a reduction in export by Ekman-driven coastal downwelling. Secondly, westerly wind anomalies at the shelf break accelerate the eastward undercurrent by inducing a barotropic meridional SSH gradient, increasing the supply of CDW to the shelf via uplifted isopycnals and the network of troughs on the shelf floor.

In austral summer (Fig. 7a), westward flow anomalies (purple anomaly) at the shelf break are indicative of a weakening in the Amundsen Sea undercurrent. CDW supply to the shelf is reduced, as shown by the

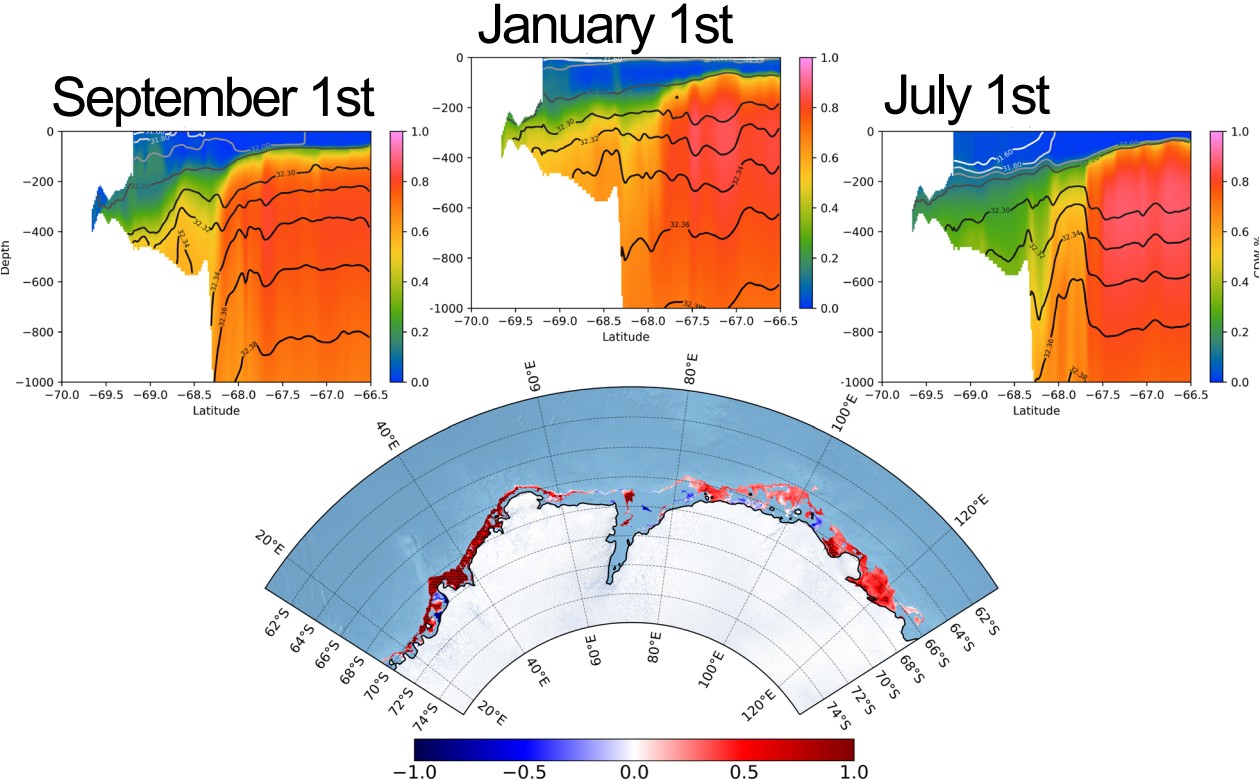

**Fig. 6 | ASF regime of seasonal CDW intrusion.** Latitude-depth cross sections at 36°E are shown (top), plotting CDW fraction and the isopycnal configuration for September 1st, January 1st and July 1st. Isopycnals flatten in austral summer, enabling CDW to flood the shelf. The monthly-mean wind-CDW correlations from Fig. 4b are shown below with a restricted East Antarctic domain. Stippling indicates regions where correlations are significant at the 90% level, assuming 10 degrees of freedom. For further significance analysis, see Fig. S4.

negative (blue anomaly) CDW fraction anomaly. In austral winter (Fig. 7b), as the atmospheric low-pressure system moves over the region, the undercurrent accelerates (green anomaly) and increases the supply of CDW to the shelf (red anomaly). This explains the winter CDW maxima along the West Antarctic continental shelf in Fig. 4a, and is consistent with some observational studies in the Amundsen Sea[57,58].

However, observational evidence of seasonality in this region is somewhat contested[59]. It should also be noted that there is a positive trend in both CDW fraction on the shelf and surface zonal wind velocity (see Fig. S6e, g) in this region. We therefore do not treat the winter maximum in CDW intrusion on to the shelf in this region with a high degree of certainty. That being said, even when de-trended, the CDW-wind correlation in this region remains significant (see Fig. S7), suggesting that the response of CDW fractions on the shelf to changes in the along-slope wind velocity at seasonal timescales remains robust. This mechanism is demonstrated in Fig. 7c, which shows the correlation between the velocity of the zonal flow at 115°W, and the bottom speed across the domain. Positive correlations throughout the domain indicate that anomalies in the zonal flow of the undercurrent in the west of the Amundsen Sea drive similar anomalies in the bottom flow speed along bathymetric troughs. The acceleration of the current drives the faster flow of CDW onto the continental shelf via this network of troughs and canyons. It is likely that there is a near zero (or at least sub-monthly) lag between the acceleration of the undercurrent and the along-canyon flows, given that both are part of the undercurrent system which accelerates quasi-synchronously as a barotropic adjustment along $f/H$ contours (where $f$ is the coriolis parameter and $H$ is depth of the water column)[60]. Correlations are largely significant at the 99% level (see stippling in Fig. 7c).

**Buoyancy-driven regimes**
**Seasonal cycle of DSW formation and export.** The Adélie Coast is the only DSW-producing region that displays considerable levels of seasonal

CDW variance on the shelf (Fig. 3b, c). The three-dimensional bathymetry of the Ross and Weddell seas, the Adélie Coast and Prydz Bay is visualised in Fig. S8 in Section S5. Unlike the Ross and Weddell seas, whose extensive shelves enable the formation of multi-year DSW reservoirs, the Adélie shelf is sufficiently narrow to sustain a seasonal cycle of DSW formation and export[13] and this is shown by the high variance values in these regions in Fig. 3d.

In the canonical cold shelf regime, the shelf remains thermally isolated from CDW due to the steep configuration of isopycnals produced by the on-shelf formation of DSW (Section "Introduction"). Cross-shelf exchange remains possible via dense overflows, but is greatly reduced in comparison to that in warm shelf regimes. Figure 8c shows that the seasonal cycle of CDW concentration at the bottom of the Adélie shelf is strongly antic-orrelated with that of DSW ($r = 0.97$). This suggests that the seasonal export of DSW from the shelf enables the intrusion of CDW along the bottom of the shelf. This is in line with the CDW intrusions that have been reported to occur at the Adélie Depression[61], and is similar to the mechanism outlined by Snow et al.[62]; the off-shelf export of DSW requires a similar on-shore flow of CDW via mass conservation. This return flow may itself be mediated by a variety of processes, such as eddies or bathymetry-steered flow. Figure 8a, b visualise this process in SOHI; plumes of CDW are shown to access the Adélie shelf through troughs in the bathymetry when high concentrations of DSW are not present at the bottom.

We acknowledge that the seasonal cycle does not close in this region. DSW fractions exhibit a near-consistent decline throughout the year. It is likely that this reflects some degree of model drift (Section "Model"), although interannual variability in forcing, interannual variability in DSW export from the Adélie Depression, or advective processes are also plausible. In order to show that the seasonal variability of CDW that we note here is not simply due to model drift, we extend the CDW time series to the maximum possible 18-months, de-trend, and re-calculate seasonal variance,

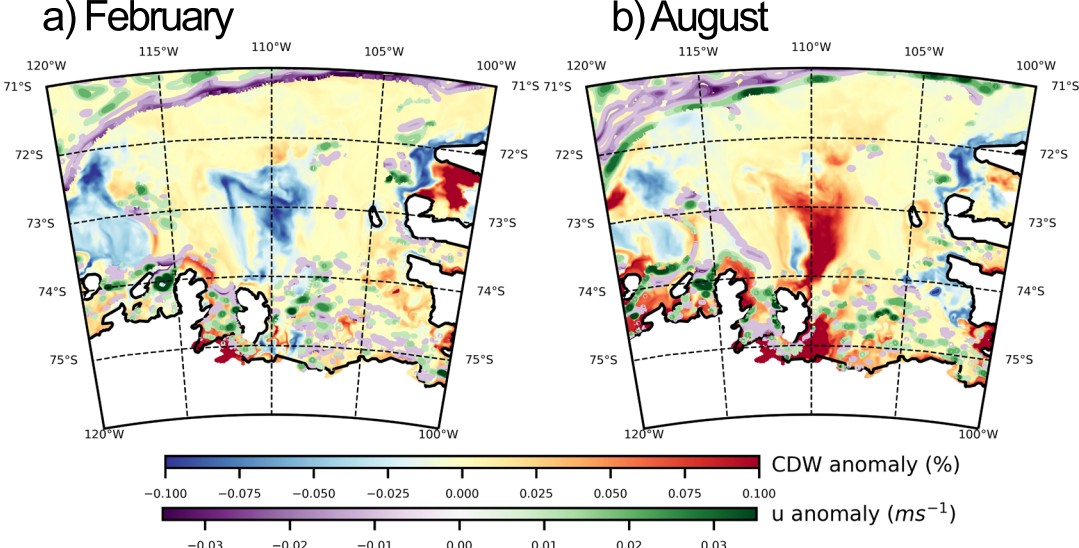

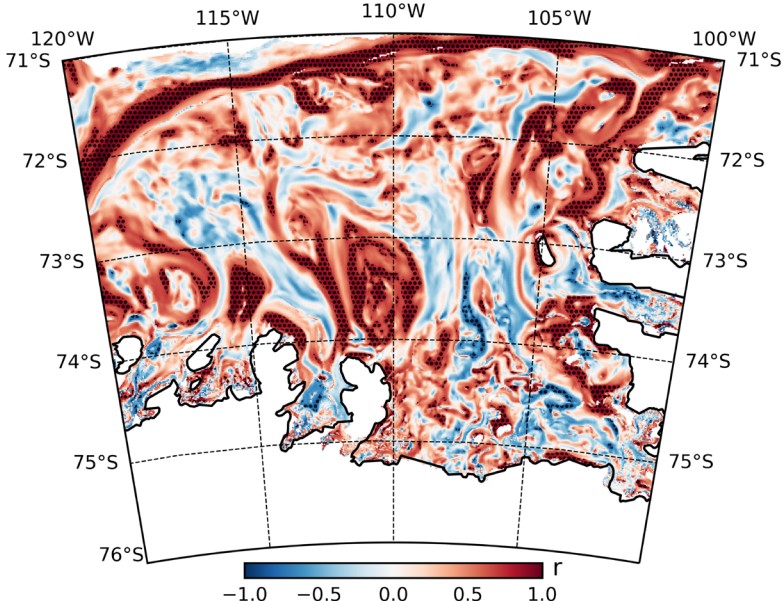

**Fig. 7 | Undercurrent regime of seasonal CDW intrusion.** CDW fraction anomaly (monthly mean - annual mean, shown in blue-red colormap) and zonal flow anomaly at 600 m (monthly mean–annual mean, shown in purple–green colormap) are shown for **a** February and **b** August. Panel **c** shows the correlation between the zonal undercurrent velocity and the bottom speed on the shelf. Undercurrent velocity is defined as the mean u-velocity across a transect along 115°W, over 71.5–71.3°S. Stippling indicates areas in which the correlation is significant at the 99% level, assuming 10 degrees of freedom.

showing that the Adélie Coast retains substantial seasonal variance (see Fig. S9). Importantly, the de-trended variability in CDW fraction in this region balances the almost entirely equal and opposite variability in de-trended DSW fraction (see Fig. S10b), such that we do not consider the strength of the relationship between these two water masses to be primarily driven by model drift. Further discussion of drift can be found in Section S6.

The Prydz Bay shelf is also narrow, with a high seasonal variance in DSW fraction (Fig. 3e). However, Figure 9 shows that this is not accompanied by strong CDW variability: CDW fractions vary by just ~4% over the year. The Prydz Bay shelf is much shallower than the Adélie shelf, such that the Prydz Bay shelf break is actually above the upper level of CDW in the water column (see Figs. 9 and S8). This is shown in Fig. 9: even when DSW is mixed away, the topographic blocking effect is such that CDW does not intrude onto the shelf at the 25% threshold. It is also visible in Fig. 3:

pathways of elevated maximum CDW concentration are visible throughout the Adélie Coast but not at Prydz Bay. Instead, the concentration of LSSW/ISW on the shelf increases via downwelling of near-surface waters that are above the shelf break. This is demonstrated by the high values of LSSW/ISW seasonal variance at Prydz Bay (see Fig. S11, Section S7). Consequently, the strength of the relationship between bottom CDW and DSW concentrations is considerably reduced compared with that in the Adélie Coast. Indeed, when the timeseries is extended to the maximum 18-month period and de-trended, the variability in DSW at Prydz Bay is balanced almost entirely by ISW (see Fig. S10c), in contrast to the Adélie Coast (where CDW fills this role, see Fig. S10b).

We therefore classify the Adélie Coast as having a buoyancy-driven regime of CDW seasonality. Variability in this regime is primarily induced by the seasonal cycle in DSW formation.

**Fig. 8 | DSW export regime of seasonal CDW intrusion at the Adélie Coast.** CDW (red) and DSW (blue) are visualised at the Adélie Coast (69–65°S, 135–145°E) for **a** September 1st and **b** March 1st. Plots show the 25% surface for CDW fraction and the 55% surface for DSW fraction. Concentrations lower than these thresholds are not shown. Seasonal export of DSW summer enables CDW intrusion along troughs. Panel c shows the mean bottom CDW and DSW fractions on the Adélie shelf throughout the year, sampled on the first day of the month. The *r*-value of 0.97 is significant at the 99% level for ≥3 degrees of freedom, and is significant at the 95% level for ≥2 degrees of freedom.

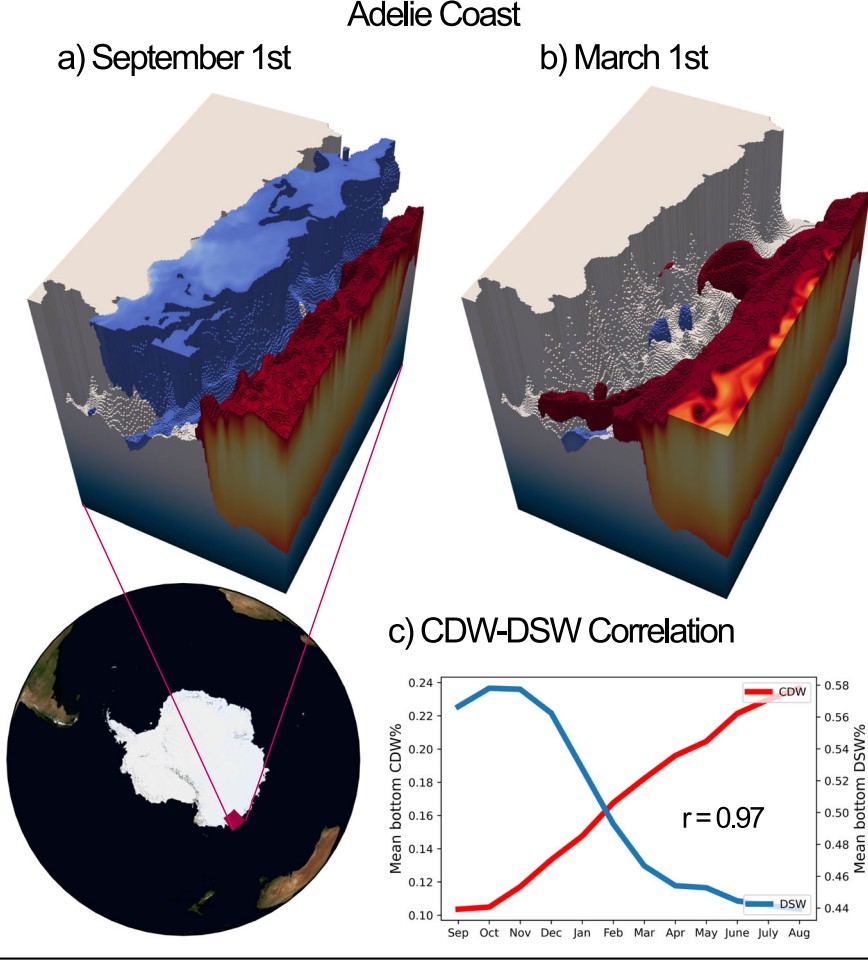

**Seasonal DSW-induced CDW inflow along canyons.** Whilst the majority of the Ross and Weddell shelf areas show low levels of seasonality due to the build up of a multi-year DSW reservoir, Fig. 3b exhibits 'fingers' of seasonal variance intruding along bathymetric canyons in these regions, which map onto areas of high maximum CDW concentrations across the season (Fig. 3a), suggesting that seasonal CDW intrusions occur along canyons in these areas.

Recent work using both models[63] and observations[64] has demonstrated that DSW outflow from the continental shelf along troughs can simultaneously induce CDW inflow along each trough's opposing flank. The mechanism is barotropic: as DSW exits via the western side of the canyon, the sea surface height is reduced on the same side. This induces a zonal pressure gradient across the canyon. Geostrophic adjustment to this pressure gradient then causes CDW to flow toward the continent on the eastern side of the canyon. We explain the seasonal intrusion of CDW along canyons with this mechanism.

Figures 10 and 11 illustrate the inflow of CDW (red) and outflow of DSW (blue) along the canyons of the Ross and Weddell seas, respectively. Figures 10c and 11c show the annual timeseries of mean CDW and DSW fractions on opposing sides of the canyons in the Ross and Weddell seas. In the Ross Sea, both the Drygalski and Joides troughs are included in the mean. The calculation for the Weddell Sea uses just the Filchner Trough. The correlations between the two timeseries are positive, and are significant at the 99% level, consistent with the barotropic mechanism outlined in Morrison et al.[63]. This finding is also congruous with a recent study by Wang et al.[65], who show that cross-shelf CDW transport in the Ross Sea is primarily driven by horizontal density gradients on the continental shelf.

Both of the 'buoyancy-forced' regimes of CDW seasonality highlighted in our work are driven by the near-shelf production and dynamics of DSW.

However, unlike in the Adélie coast, canyon-driven intrusions in the Ross and Weddell seas occur during or soon after periods of maximum DSW formation.

## Discussion and conclusions

In this work, we identify four distinct regimes in the seasonality of CDW intrusion and the properties of water masses on the circumpolar Antarctic continental shelf. These are classified into two wind-driven (Section "Wind-driven regimens") and two buoyancy-driven (Section "Buoyancy-driven regimens") regimes (Fig. 5). Wind-forced regimes are shown to entail an ASF mechanism (East Antarctica) and an undercurrent mechanism (West Antarctica). We use a metric for ASF detection based on the meridional gradient of SSH (Fig. 1), which naturally distinguishes between these two regimes. Buoyancy-forced regimes are governed by either DSW formation/export or canyon inflow mechanisms. These regimes are characterised by distinct topographic configurations of the shelves. Some shelves are found to exhibit little-to-no seasonality (Section "Results"), and this is attributed to either the build up of a multi-annual DSW reservoir (Weddell and Ross seas) or the pronounced shallowness of the shelf (Prydz Bay).

Our results indicate that there are three primary factors controlling seasonality of CDW intrusion onto the shelf: wind forcing, buoyancy forcing, and bathymetric configuration. The four shelf regimes of CDW seasonality are a result of a complex interplay between these three factors. Specifically, we find that the buoyancy forcing is the dominant factor determining the seasonality regime. Where no DSW is formed, CDW concentrations on the shelf are sensitive to changes in wind forcing (Fig. 6). Where DSW is produced, the seasonal cycle of CDW on the shelf is largely decoupled from wind forcing (Fig. 4b). In this scenario, the production of DSW and associated vertical convection alter the isopycnal configuration to

**Fig. 9 | A lack of seasonal CDW intrusion onto the Prydz Bay Shelf.** CDW (red) and DSW (blue) are visualised at Prydz Bay (69–66°S, 60–70°E) for **a** September 1st and **b** February 1st. Plots show the 25% surface for CDW fraction and the 55% surface for DSW fraction. Concentrations lower than these thresholds are not shown. Seasonal export of DSW during late austral summer is not associated with CDW intrusions at the 25% threshold. Panel **c** shows the mean bottom CDW and DSW fractions on the Prydz Bay shelf throughout the season, sampled on the first day of the month. The *r*-value of 0.59 is not significant at the 99% level.

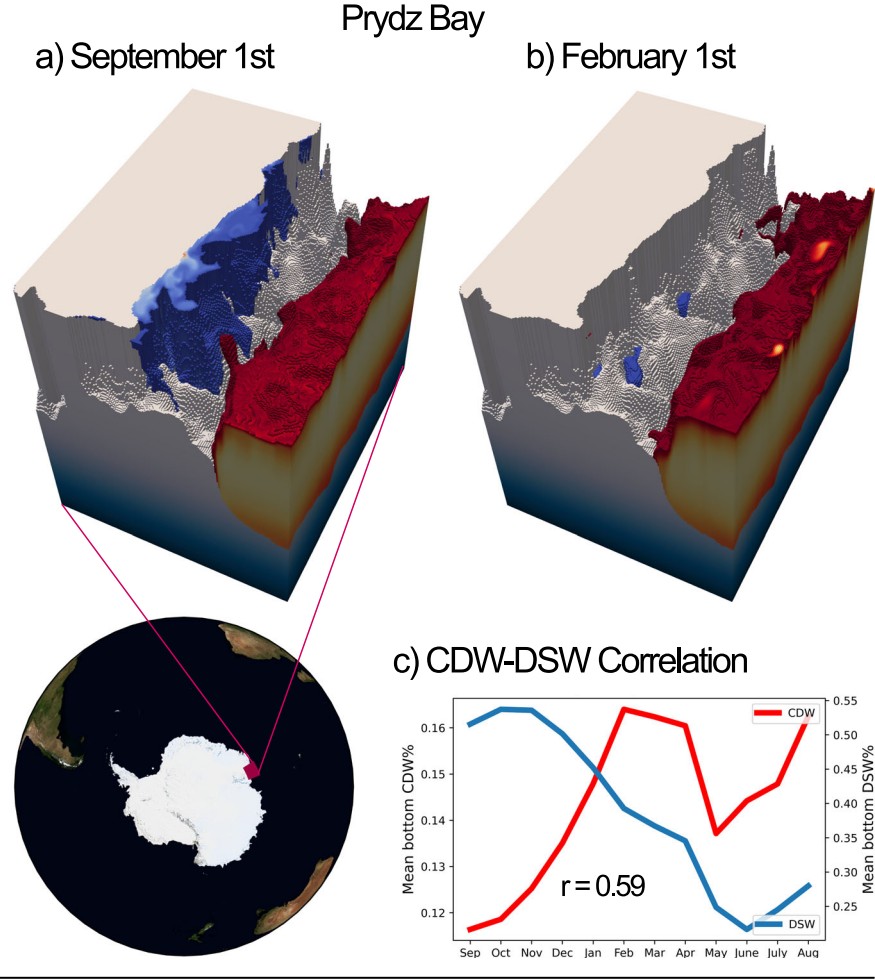

produce a strong shelf-break front, which opposes shoreward Ekman transport and downwelling of fresh surface waters. In the case of the Adélie Coast, wind-CDW fraction correlations are negative, and the minimum in DSW concentration occurs at roughly the same time as the strongest easterly winds. Within DSW-producing regions, bathymetry is the leading-order determinant of whether a shelf exhibits seasonality, and by which mechanism.

Traditional Antarctic shelf regime classifications are based on time-mean ocean conditions and shelf geometry. Here, we propose that these classifications can be developed to account for the distinct factors that regulate the seasonality in the water mass configuration of the shelves, which is arguably as important to Antarctic ice sheet melting as the shelves' annual-mean conditions. Cold shelves exhibit buoyancy-driven seasonality (of either mechanism in Section "Buoyancy-driven regimens") or no substantial seasonality. Fresh-shelf seasonality is either governed by the wind-driven ASF mechanism (Section "Wind-driven regimens") or by advection (Supplementary Information, Section S2). Warm shelves either exhibit little-to-no seasonality, or display seasonal variability that is driven by variability in a wind-driven shelf-break undercurrent (Section "Wind-driven regimens").

For some regions along the Antarctic margin, the prominence of seasonality may increase in the remainder of this century. The intensified seasonality in Antarctic easterly winds that is evident in reanalysis data over the past 40 years[39] suggests that the ASF mechanism may be becoming more pronounced and may continue to become so in the future. Given that DSW (and AABW) is predicted by many studies to decline with positive buoyancy anomalies linked to surface warming and accelerated ice shelf melting[10,44,66,67], our results suggest that DSW-producing shelves with

buoyancy-forced seasonality may become increasingly sensitive to CDW influx from seasonal wind forcing. It is possible that transitions from buoyancy-driven to wind-driven regimes may already be underway in some of these regions[53], and that such transitions may provide important positive feedbacks in the future climatic evolution of Antarctica and the stability of the Antarctic ice sheet[68,69].

Looking forward, we argue that the current set of operational coupled climate models may be particularly poorly suited to represent the seasonality of CDW intrusion onto the Antarctic continental shelf. These results highlight that the production and export of DSW from the shelf is key to the dynamics of CDW intrusion, meaning that coarse-resolution models that struggle to resolve DSW formation will not capture the complexity of seasonal CDW intrusion. Resolution of bathymetry is also likely to be a limiting factor on projection skill, since we show that continental and ice shelf geometry—including canyons and troughs likely to be poorly represented in coupled climate models—is of leading-order importance in determining the shelf's seasonality regime.

## Model and methods
### Model
There are several key limitations of coupled general circulation model projections of changes to Antarctic continental shelf and sub-ice shelf water mass processes. Coarse-resolution models struggle to resolve DSW and AABW formation[45–48]. Only recently have studies succeeded in running future simulations to at least mid-century with a sufficiently high resolution to capture such processes[44]. Generally ranging in resolution between 1/4 and 1 degree (~30–110 km), climate models are also too coarse to resolve most mesoscale and submesoscale eddy variability. Only at resolutions on the

**Fig. 10 | Canyon regime of seasonal CDW intrusion in the Ross Sea.** CDW (red) and DSW (blue) are visualised on the continental shelf of the Ross Sea (77.5–71°S, 163–180°E) for **a** September 1st and **b** February 1st. Plots show the 48% surface for CDW fraction and the 68% surface for DSW fraction. Concentrations lower than these thresholds are not shown. Some of the variability at shallower depths has been removed in order to better view canyon processes. CDW inflow along the eastern side of canyons is shown to be concurrent with DSW outflow along the opposing side. Panel **c** shows the annual time-series of CDW and DSW fractions (sampled on the first day of the month) along the eastern and western edges of the canyons, respectively. The three-dimensional averaging area for CDW is defined as any point in which CDW fraction exceeds 45% on the 1st November 2005, within the box: 72. 5°S < latitude < 75°S, 168°E < longitude < 180°E, and 200 m < Depth < 900 m. The DSW averaging volume is defined as any point in which DSW fraction exceeds 70% within the same box on the same date. As can be seen from **a**, these criteria produce averaging areas on eastern (CDW) and western (DSW) sides of the canyon. This includes both the Drygalski and Joides troughs, visible in **a** and **b**. The *r*-value of 0.8 of this relationship is significant at the 99% level for ≥7 degrees of freedom, and is significant at the 95% level for ≥5 degrees of freedom.

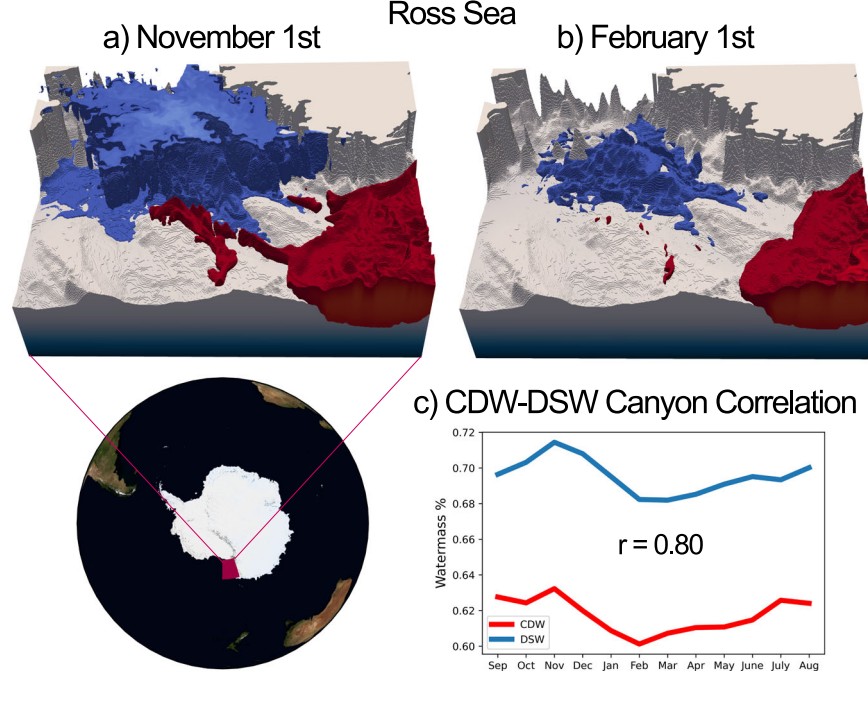

**Ross Sea**

a) November 1st
b) February 1st

c) CDW-DSW Canyon Correlation

r = 0.80

**Fig. 11 | Canyon regime of seasonal CDW intrusion in the Weddell Sea.** CDW (red) and DSW (blue) are visualised at the Weddell Sea (81–72°S, 45–30°W) for **a** December 1st and **b** July 1st. Plots show the 40% surface for CDW fraction and the 48% surface for DSW fraction. Concentrations lower than these thresholds are not shown. Some of the variability at shallower depths has been removed in order to better view canyon processes. CDW inflow along the eastern side of the Filchner Trough is shown to be concurrent with DSW outflow along the opposing side. Panel **c** shows the annual time-series of CDW and DSW fractions (sampled on the first day of the month) along the eastern and western edges of the canyon, respectively. The three-dimensional averaging area for CDW is defined as any point in which CDW fraction exceeds 40% on the 1st December 2005, within the range: 75°S < latitude < 81°S and 30°W < longitude < 45°W. The DSW averaging volume is defined as any point in which DSW fraction exceeds 50% on the same date within the same latitude-longitude range, with the added constraint that Depth <600 m to exclude DSW on the ridge of the canyons. As can be seen from **a**, these criteria produce averaging areas on eastern (CDW) and western (DSW) sides of the canyon. The *r*-value of 0.8 of this relationship is significant at the 99% level for ≥7 degrees of freedom, and is significant at the 95% level for ≥5 degrees of freedom.

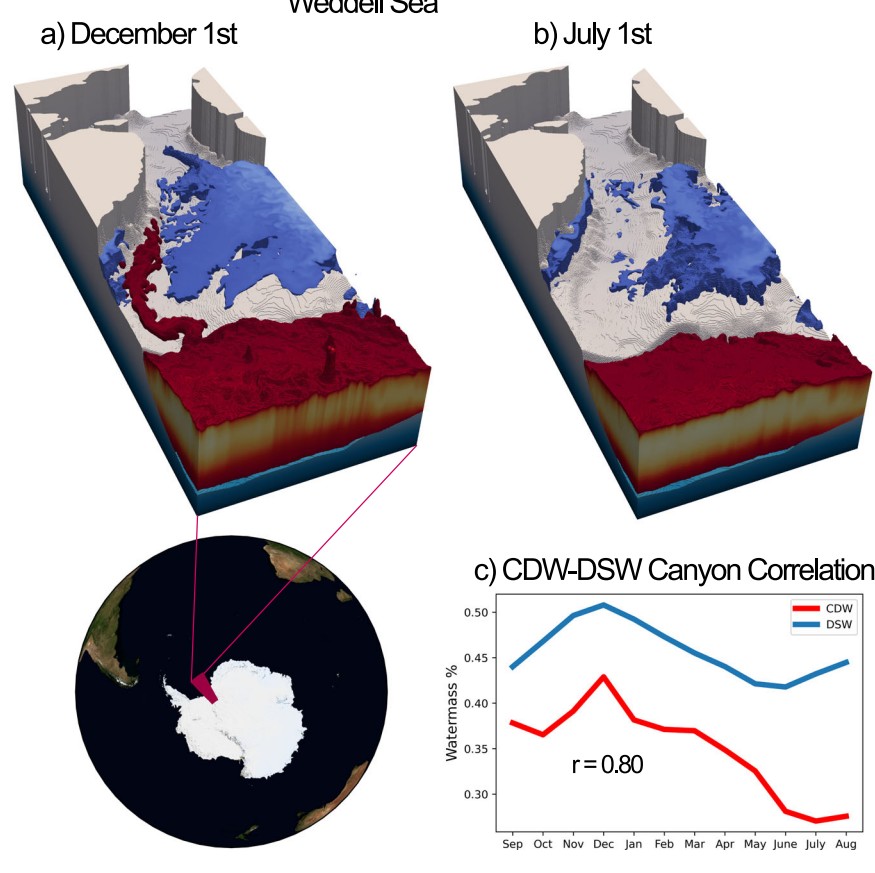

**Weddell Sea**

a) December 1st
b) July 1st

c) CDW-DSW Canyon Correlation

r = 0.80

order of 1/10° (~12 km) are the largest baroclinic eddies resolved. This means that these models cannot directly simulate the eddy fluxes known to shape some of the cross-slope CDW incursions. The majority of coupled models also do not include ice shelf cavities, and their representation of topography, including the continental shelf and slope, is necessarily constrained by horizontal and vertical resolution. This means that the channelling processes and grounding line geometry that control CDW exposure to the shelf are similarly limited.

To overcome these limitations, we use a high-resolution, eddy-resolving formulation of MITgcm (SOHI) with realistic bathymetry, sub-ice shelf cavities, and a coupled sea-ice model. It has 1/24th degree horizontal grid spacing from latitude 85.5°S to 40°S, telescoping to 1/12th at 40°S. It remains at 1/12th grid spacing until the northern boundary at latitude 0.6°S. The model employs a Mercator projection, so this resolution translates to approximately 1.2 km at 75°S. The model has 225 vertical levels. Tidal forcing is prescribed at the surface via a pressure loading field derived using astronomical inputs from NASA's Navigation and Ancillary Information Facility and the NASA SPICE toolkit[70]. Barotropic tides are also prescribed at the open boundary from TPXO7[71]. Temperature, salinity and velocity are further prescribed at the open boundary from ECCO (estimating the circulation and climate of the ocean) version 4 release 4. Atmospheric forcing for both the ocean and sea ice models is provided by ERA5 hourly surface winds, heat, and freshwater fluxes. The bathymetry is the 2020 GEBCO grid, combined with BedMachine v1.39 cavity representation. Diagnostics are archived as 6-hourly averages. Further information on the SOHI configuration, as well as additional validation, can be found in Dinh et al.[72].

SOHI is initialised from SOSE[73]. SOSE has 1/6th degree resolution and 42 vertical levels. It assimilates a suite of observational data, including Argo float profiles, CTD sections, seal-acquired hydrographic data, and an array of altimetry products. Whilst these sources of data are relatively numerous throughout the Southern Ocean, we note that, on the continental shelf, observational data is more scarce and the constraints are likely to be weaker. For the full model description of SOSE, see Mazloff et al.[73]. Given the close relationship between the two models, we begin the analysis by comparing the SOHI solution with SOSE to highlight the addition of features and emergence of new dynamics that allow for the formation of DSW and its seasonality (Section "Results").

We consider a single model year from September 2005 to August 2006; the only full model year currently available for SOHI. The justification for this choice comes from the parent model, SOSE, for which the Argo float data that it assimilates begin to become available at this time at sufficient resolution to constrain the model. The September start month allows for an 8-month spin-up time from the point of nesting, after an initial 1-year spin-up at 1/12th degree resolution. All data used in this study occur after the spin-up. There is an additional 6 months of data spanning from September 2006 to March 2007, which we primarily use to analyse the extent of model drift in Section S6.

SOHI is an ambitious project in the early stages of conception, with high associated costs of running. At the time of writing, 18 months of data were available, with the model running at a slower pace than real time. We anticipate that SOHI will soon become a multi-year product, but at this stage we are limited to analysis of a single seasonal cycle, which we believe is sufficient to identify the large-scale regimes of seasonal variability along the Antarctic margins. At some stage, analysis of how these regimes vary with time will become possible in SOHI. However, for now, we contend that a single year is a necessary compromise to enable us to reach sufficiently high resolution to resolve the key finescale processes known to be important in the region.

The spin-up length is sufficient to stabilise the kinetic energy in the model, although there is a suggestion of drift in $\theta$ and $S$ in some regions on the shelf. Most notably, the cycle of DSW formation on the Adélie Coast appears as a near-consistent decline across the model year. Other regions, such as Prydz Bay or Queen Maud Land, for example, show a more complete closure of the water mass seasonal cycle. It is possible that the model is yet to fully equilibrate in some areas of the Antarctic shelf, although some of

this signal could also be attributable to interannual trends in the wind forcing or advection. We provide an in-depth analysis of model drift in Section S6.

## Verification against observations

SOHI is a downscaled version of SOSE and is thereby constrained by the aforementioned extensive observational data that SOSE assimilates[73]. We verify the near-shelf conditions of the model with Marine Animals Exploring the Oceans Pole to Pole (MEOP) seal-aquired hydrographic data. Figure 2 shows temperature–salinity (T–S) bivariate histograms as contours for eleven near-shelf regions in which MEOP data density is relatively high. Seal data measurements are geo-located in the model, and the values for the nearest grid point and day are taken as a sample. Due to the relative scarcity of MEOP data, all MEOP years are included in the comparison with the 2005/6 model run. This is because we find a high risk of oversampling below ~5000 data points (see Figs. S12 and S13), although this does mean that interannual variability is unaccounted for. Most regions exhibit good model-observation agreement in mean T–S conditions. In regions of poor agreement, such as the Bellingshausen and Amundsen seas, below-400 m agreement is markedly improved relative to the full depth range (see Fig. S14), suggesting that the discrepancy arises within the mixed layer or is related to the mixed layer depth. Moreover, in these regions, the model T–S distribution is more consistent with known T–S ranges within the West Antarctic continental shelf: the seals appear not to sample the CDW ($\theta \geq 0°C$, $S > 34.5$) known to be present in the Amundsen and Bellingshausen seas in high concentrations, suggesting that data are sampled primarily from the fresh and cold mixed layer. Given that our analysis is focused on processes at or close to the seabed of the continental shelf, this does not pose a problem for our investigation. It is worth noting that most regions may not be sufficiently well-sampled at depth to provide robust comparisons in hydrographic properties below 400 m (see Figs. S14 and S13). That being said, it is possible that there is a warm bias on/near the shelf at Knox Coast below 400 m (Fig. S14, consistent with Dinh et al.[72]). Further verification of SOHI's near-shelf hydrography, along with standard errors, is provided in Section S8 in the Supplementary Information. For further validation, including an assessment of the accuracy of sea ice cover, see Dinh et al.[72].

Despite any sampling issues, agreement between the modelled and MEOP seasonal T–S cycles is strong (see Fig. S15). MEOP data sparsity across the seasons prevents direct comparison of the full cycle in all but a few regions (such as the Bellingshausen Sea, shown in Fig. S15e). Instead, monthly comparisons are made across all shelf regions, and we find good coherence between monthly T-S values at all shelves studied (see Figs. S15a and S15b). Agreement is improved even further by excluding low-sampling shelf months: salinity r-values increase if months with less than 20,000 data points are excluded (see Figs. S15c and S15d), suggesting that undersampling by MEOP data limits apparent model skill. Indeed, both temperature and salinity r-values converge to near 1 as the sample size increases (see Fig. S15f).

## Water mass classification: machinery and source water type selection

We classify water masses in the model using the least-squares machinery outlined in Wunsch[74]. This is similar to a classical Optimum Multi-parameter (OMP) analysis. We select values of potential temperature and salinity to represent four unmixed Source Water Types outlined below. It is assumed that both potential temperature and salinity undergo the same mixing processes with identical mixing coefficients. The analysis is constrained in that all water mass fractions should add up to approximately 1, which is a statement of mass conservation. Deviations from 1 are allowed within prescribed uncertainty, recognising that other source waters exist that are not being modelled. Both potential temperature and salinity are assumed here to be conservative. We acknowledge that, especially near the surface and the ice shelves/glaciers, there may be non-conservative buoyancy fluxes that are only accounted for by the analysis via the modest relaxation of mass conservation. We assume an equal a priori water mass

likelihood and impose a stronger constraint on the mass conservation equation. The full set of equations, including weighting and normalisation practices, are set out in Section S1.3 in the Supplementary Information. Further information relating to residuals and negatives in the solutions can be found in Section S8.1 and in Figs. S16 and S17.

Four water masses are selected for classification: CDW, DSW, LSSW and ISW. Source water end member values are chosen from known T-S ranges and tailored to the extremes of the SOHI parameter space (see Fig. S18). CDW and DSW are chosen as the warm/salty and cold/salty edges of the parameter space, respectively. We opt to select a single CDW source water for our analysis, recognising that, as CDW mixes with waters on the shelf, the CDW label also encompasses what would often be classified as modified CDW. We select a LSSW end member to represent the low-salinity cold shelf water associated with Ekman-driven downwelling of the surface layer. This is chosen as the midpoint of the narrow T–S range given by Schodlok et al.[2]. We define a fourth source water type, ISW, as the cold and fresh extreme of the distribution, describing the water that is produced by ice shelf melting. However, we acknowledge that this is often not the true ISW found in observations[75]. The modelled T–S distributions (see Fig. S18) show that the model does not produce the sub-freezing point supercooled temperatures associated with refreezing and the formation of ISW. This issue is common within the ice-shelf cavities of numerical models of the Southern Ocean[76]. We also acknowledge that some of what we classify as ISW may contain the capped cold fresh surface water known as Winter Water. The distinction between these is not important to our analysis, and further discussion of where this occurs can be found in Section S9.1 in the Supplementary Information. We run the analysis in both SOHI and SOSE, using identical machinery and source water end members.

Due to computational constraints, the circumpolar water mass analysis uses twelve monthly-mean timesteps, unless otherwise stated. In cases in which we use a daily snapshot, we calculate a 24-h average period to remove the influence of most tidal constituents. ERA5 wind fields are also analysed as monthly averages.

## Data availability

The MEOP data that we used to validate SOHI is available here https://meop.net/database/. SOSE fields are available here: https://climatedataguide.ucar.edu/climate-data/southern-ocean-state-estimate-sose. ERA5 wind fields were downloaded from Copernicus Climate Data Store (https://cds.climate.copernicus.eu/).

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

## Acknowledgements

J.A.L. acknowledges EPSRC doctoral training grant. M.R.M. acknowledges support from NSF awards OCE-1924388, OPP-2332379, and OPP-2319829, and from NASA awards 80NSSC22K0387 and 80NSSC20K1076. A.C.N.G. acknowledges U.K. Research and Innovation guarantee funding for a European Research Council Advanced Grant (EP/X025136/1). A.M. acknowledges ONR Grant N00014-22-1-2082.

## Author contributions

J.A.L. led the analyses and writing, M.R.M. led the modelling effort and contributed to development of various methodologies, and A.C.N.G. contributed to devising the framework of the analyses. J.A.L., A.M., M.R.M., A.C.N.G., and M.S. contributed to interpretation of results and the writing of the paper.

## Competing interests

The authors declare no competing interests.
