## [Transparent Peer Review File · Communications Earth & Environment]

Seasonal regimes of warm Circumpolar Deep Water intrusion toward Antarctic ice shelves.

Corresponding Author: Mr Joshua Lanham

Version 0:

Decision Letter:

Dear Mr Lanham,

Your manuscript titled "Seasonality of Intrusion of Warm Circumpolar Deep Water onto Antarctic Shelves" has now been seen by 3 reviewers, and we include their comments at the end of this message. They find your work of interest, but some important points are raised. We are interested in the possibility of publishing your study in Communications Earth & Environment, but would like to consider your responses to these concerns and assess a revised manuscript before we make a final decision on publication. Specifically, we ask you to:

1. Fully explain the choice of using a single day per month to diagnose seasonality and provide more robust validation for seasonal variation as well as water mass analysis.
2. Clarify your physical reasoning for analysis and compellingly demonstrate the mechanisms driving on-shelf CDW intrusions.

We therefore invite you to revise and resubmit your manuscript, along with a point-by-point response that takes into account the points raised. Please highlight all changes in the manuscript text file.

Please submit your point-by-point responses as a separate file, distinct from your cover letter where you can add responses to the Editors' comments that you do not want to be made available to the reviewers. Word files are preferred.

Important: The response to reviewers must not include any figures, tables or graphs. If you wish to respond to the reviewer reports with additional data in one of these formats, please add them to the main article or Supplementary Information, and refer to them in the rebuttal. Due to current technical limitations, any figures, tables, or graphs embedded in your rebuttal will not be included in the peer review file, if published.

Please use the following link to submit your revised manuscript, point-by-point response to the referees' comments (which should be in a separate document to any cover letter), a tracked-changes version of the manuscript (as a PDF file) and the completed checklist:

Link Redacted

We hope to receive your revised paper within six weeks; please let us know if you aren't able to submit it within this time so that we can discuss how best to proceed. If we don't hear from you, and the revision process takes significantly longer, we may close your file. In this event, we will still be happy to reconsider your paper at a later date, as long as nothing similar has been accepted for publication at Communications Earth & Environment or published elsewhere in the meantime.

Please do not hesitate to contact us if you have any questions or would like to discuss these revisions further. We look

forward to seeing the revised manuscript and thank you for the opportunity to review your work.

Best regards,

Dr Alireza Bahadori
Associate Editor
Communications Earth & Environment

EDITORIAL POLICIES AND FORMATTING

Editorial Policy: [Policy requirements](https://www.nature.com/documents/nr-editorial-policy-checklist.pdf) (Download the link to your computer as a PDF.)

- Behavioural and social science
- Ecological, evolutionary & environmental sciences
- Life sciences

<https://www.nature.com/documents/nr-reporting-summary.zip>

Furthermore, please align your manuscript with our format requirements, which are summarized on the following checklist: [Communications Earth & Environment formatting checklist](https://www.nature.com/documents/commsj-phys-style-formatting-checklist-article.pdf)

and also in our style and formatting guide [Communications Earth & Environment formatting guide](https://www.nature.com/documents/commsj-phys-style-formatting-guide-accept.pdf).

***** DATA:** Communications Earth & Environment endorses the principles of the Enabling FAIR data project (<http://www.copdess.org/enabling-fair-data-project/>). We ask authors to make the data that support their conclusions available in permanent, publically accessible data repositories. (Please contact the editor if you are unable to make your data available).

All Communications Earth & Environment manuscripts must include a section titled "Data Availability" at the end of the Methods section or main text (if no Methods). More information on this policy, is available at <http://www.nature.com/authors/policies/data/data-availability-statements-data-citations.pdf>.

If a community resource is unavailable, data can be submitted to generalist repositories such as [figshare](https://figshare.com/) or [Dryad Digital Repository](http://datadryad.org/). Please provide a unique identifier for the data (for example a DOI or a permanent URL) in the data availability statement, if possible. If the repository does not provide identifiers, we encourage authors to supply the search terms that will return the data. For data that have been obtained from publically available sources, please provide a URL and the specific data product name in the data availability statement. Data with a DOI should be further cited in the methods reference section.

REVIEWER COMMENTS:

Reviewer #1 (Remarks to the Author):

The manuscript analyses one year of the SOHI model and identifies four regimes of seasonal CDW intrusion onto the Antarctic continental shelf. The watermass fraction analysis is a useful framework for untangling the complex dynamics of this region. The manuscript is a significant contribution to the field, and extends on the usual time-mean regime understanding of the Antarctic margins. The manuscript is well written with some very nice visualisations. I do however have some concerns, outlined further below, about the influence on the results of the large model drift and the choice of using a single day per month to diagnose seasonality. In addition, many regions in the correlation maps have low significance. Therefore I recommend major revisions. However, I did really enjoy reading this work and I look forward to reviewing a revised manuscript.

Major comments:

1. It would be helpful to add regions into the abstract for each of the 4 regimes, for those time-poor scientists who will skim the paper. e.g. regime 1 (East Antarctica), regime 2 (West Antarctica), regime 3 (Adelie coast), regime 4 (Ross/Weddell Seas).
2. Line 84: "The response of shelf regimes to climate change is likely to hinge upon changes in the atmospheric circulation around Antarctica". Actually, I disagree with this statement. Yes, wind changes will undoubtedly play a role, but models projections (e.g. Beadling et al. 2022; Li et al. 2023) have shown that future change around Antarctica is likely to be dominated by meltwater forcing changes, not atmospheric forcing changes. In particular, regimes 3 and 4 that depend on DSW will likely be more dependent on changes in meltwater forcing. I suggest adding an additional paragraph after this one, focusing on the impact of potential meltwater forcing change, in addition to this paragraph on potential wind forcing change.
3. An 8-month spinup period seems very short. Lines 160-165 hint that the drift in some regions is large (also as confirmed by Figures 8c and 9c, and even CDW in Figure 11c), but other regions are ok. The manuscript really needs to include validation to convince the readers that this short spinup is sufficient and that the seasonal variation that this paper focuses on is indeed seasonality and not drift. Options could be:
 - a) A plot showing a map of the magnitude of the drift in CDW fraction over the 1 year period, compared with the magnitude of the detrended seasonal cycle of CDW fraction at each location.
 - b) Time series from the start of the spinup to the end of the extended 2 month period after the 12 months analysed. These plots would help the reader understand in which regions the results are likely to be robust, and which regions need to be considered more carefully.If these plots show that the drift is larger than the seasonal cycle, then the output could be detrended before analysis.
4. I understand the authors' reasoning behind using full depth MEOP data in the comparison in Figure 2, because there is more limited observational data available at depth. However, if possible, it would be a more appropriate and useful validation to only compare against MEOP in the deeper ocean (e.g. below 400m or where there is CDW), given that the focus of the paper is on CDW. I would be interested in seeing Figure S1 calculated only for a deeper subset (e.g. below 400m) of the MEOP data. I'm guessing that it may converge faster (i.e. a smaller number of data points may be acceptable), because there is likely to be less variation in T and S below the mixed layer. If this is the case, it would be a nice addition to add a second MEOP comparison figure into the main manuscript - e.g. add W. Ross and Knox T/S sub panels to Figure S2 and move it to Figure 2 (or even replace Figure 1?) in the main manuscript.
5. Line 227: I really like this water mass fraction analysis! However, it seems to me that it would be better to use monthly averages for the water mass analysis, rather than a single daily average each month. Monthly averages would be more representative of the whole month. Especially for the watermass variance shown in Figure 3, the high variance here could just be the result of eddies in particular daily averages rather than high seasonal variance. Also for Figure 5b, the correlation is between the monthly wind stress and the daily watermass fraction. It would make much more sense here to use the same time period of averaging for both quantities before correlating. I don't understand why using daily averages would be computationally easier than using monthly averages. Once monthly averages are computed, then there would have the same number of data points as using a single day each month.
6. Line 243: "The lack of cavities means that SOSE does not represent significant areas of DSW production in the Ross and Weddell seas." I disagree with this logic. Other models (e.g. ACCESS-OM2-01, Moorman et al. 2020) produce DSW without ice shelf cavities. You can also see in Figure 3c that the DSW fraction is highest over the open continental shelf regions where winds and atmospheric cooling are high, not under cavities (except in the western Ross, where I'm guessing the DSW forms on the open continental shelf and flows under the cavity).
7. It would be a useful addition to add a map similar to Figure 5, but with all locations with high CDW fractions (>0.3?) shown (not just those with high variance). There are areas with strong CDW intrusions (e.g. Drygalski trough, Amery) which don't show up in Figure 5. Is the seasonality in these regions really close to zero? I'm wondering if the current variance metric could be dominated by high frequency (eddy) variations in some locations, e.g. variance is high along the shelf break around much of East Antarctica, and the peak months are quite spatially inhomogeneous in many regions (east of Amery, George VI). Such an additional map figure could go in the supplementary if needed. If the true seasonal variance in these other locations is indeed weak, then this is not needed.
8. There are many areas of the correlations shown in Figure 5b that are not significant to the 90% level. I suggest only showing areas on Figure 5 that are significant, or adding stippling as done in Figure 7c. If non-significant regions are

masked, then the current map could be added to Figure S6 instead.

9. I really like the regime classification in Figure 4. However, I think it would be easier to interpret (and more likely to be reused in other people's talks etc) if the schematics were enlarged and had a little more detail. e.g. For the ASF regime it would be useful to show strong winds for the top panel and weak winds for the lower panel. I struggle to interpret what the undercurrent regime schematic is showing - CDW moving upwards in the water column and travelling westward along the shelf break? How is that relevant for intrusions onto the shelf? Is the lower line the undercurrent? But it's travelling westward not eastward? If it is the undercurrent, it could be labelled. For the Canyon Regime, is the upper line SSH? Or an isopycnal? This could be labelled.

10. Caption of Figure 6: "These correlations are significant at the 99% level over 20-60°E and at the 90% level over 80-130°E (see Figure S6)." This does not seem consistent with what is shown in Figure S6. From Figure S6c, it looks like most of 20-60°E is closer to 95% significance, with only limited points in that longitude range being significant at 99%. Between 80-130°E, it looks to me like most points are not even significant at 90% (Figure S6a). As per my comment above for Figure 5, I suggest limiting what is shown in Figure 6 to only include points that are deemed significant; or add stippling.

11. Lines 359-362: "Figure 8c shows that the seasonal cycle of CDW concentration at the bottom of the Adelie shelf is strongly anticorrelated with that of DSW ($r^2 = 0.93$). This suggests that the seasonal export of DSW from the shelf enables the intrusion of CDW along the bottom of the shelf." I strongly disagree with this conclusion. As the authors acknowledge as a possibility later in this paragraph, I think Figure 8c (also 9c) predominantly shows model drift, not seasonality. It looks to me like the model is not producing as much DSW compared to what is present in the initial conditions, so that the fraction of DSW is tending downwards (both in Adelie and Prydz). As the DSW fraction tends downwards, this watermass is instead being replaced by CDW. Figure S10 shows that there is some seasonal production of DSW, but the annual production magnitude looks to be much smaller than the model drift. I find it problematic that the reason the Adelie coast is showing up with high variance in Figure 3b,d is likely just due to model drift and not seasonal variability. As suggested above, perhaps this could be addressed by detrending each location first before calculating the seasonality/variance shown in Figures 3 and 5.

Line 369: "It is possible that this could result from model drift". This seems like the most logical conclusion to draw. It would be useful to increase the length of the time series in Figure S10 to include the first 8 months of the spinup to clarify this.

Line 372-373: "Regardless, the strength of the relationship between DSW and CDW on this shelf is of primary mechanistic importance." I disagree that you can conclude anything about mechanisms from Figure 8c if you cannot rule out the possibility that it is just drift.

12. Line 378: "even when DSW is exported". I have not seen any evidence in the rest of the paper that DSW is exported to the abyss in this model. From Figure 3c it looks to me like there is no connection between the DSW watermass on the shelf and in the abyss (e.g. in the Ross and Weddell). Instead there is a ring of CDW at the ocean floor at the base of the continental slope. I would have thought that even after 1 year, you might see overflows reaching deeper down the slope if they are present in the model. To make a statement like this would require showing that DSW is exported off the shelf (to the abyss or mid depths?) rather than just mixed away.

13. Lines 387-388: "the seasonal cycle in DSW formation, which reaches a maximum in winter". I don't see this in Figure 9c? To me it looks like the DSW fraction is minimum in winter (June/July)?

14. I like the time series in Figures 8-11, because they help show the magnitude of CDW seasonal variability, and put this into the context of model drift. It would be nice to add similar time series (perhaps with LSSW instead of DSW) to Figures 6 and 7 for the East and West Antarctic regions.

Minor comments:

- Line 28: "affected" not "effected"?

- Paragraph at line 84: It would be helpful to also cite Neme et al. 2022 here, which analyses future projections of change in the easterly winds close to Antarctica.

- Line 123: "GCMs also do not include ice shelf cavities." This is a bit of a sweeping generalisation. e.g. What about UKESM (Smith et al. 2021)? Suggest adding "The majority of..." or similar.

- Line 135: Is sea ice forcing included in the atmospheric forcing that's applied? Best to specify this here.

- Figure 2: The outside subpanels need 'Temperature' and 'Salinity' labels on the axes.

- Caption of Figure 2: "Due to computational constraints, these plots include just the first 6 months of SOHI data (September - March)". I am confused by this statement. Does this mean that no MEOP data from the months April - October is included in the comparison, because it is not possible to sample the model in those months? I also don't understand why this is computationally infeasible.

- I suggest reducing the use of acronyms. Acronyms reduce the readability of the paper. In particular, I don't think GCM, SWT

and WMT are needed.

- Figure S4 is not referenced in the main text.

- Figure 3a - Is this SOSE or SOHI? It would help to add the model to the title as is done for panels c and e. Or else remove the SOHI label from panel c if all other panels except e refer to SOHI.

- The ordering of the Supplementary figures does not match the order they are called in the main manuscript. e.g. Figure S7 is called before Figure S6.

- Figure 5 is discussed (line 251) before Figure 4 is discussed (line 280). I suggest swapping the order of these figures.

- The placement of the colorbar in Figure 5a covers the longitude labels and is too close to panel b.

- Line 402: "green" should be "blue".

- Line 413-414: "Surface WMT is initiated in March in the Ross Sea, and in April in the Weddell Sea" (and similar statements throughout this paragraph). Does this refer to the seasonality from Schmidt et al. 2023 or SOHI? I would expect this to be model dependent, because the calculation depends on model surface density (and therefore model advection timescales etc). Therefore I don't think it's a relevant comparison to use the timing directly from Schmidt et al. 2023 if that is what is done here. However this is straightforward calculation to redo from the SOHI model if required, only requiring surface fluxes/density.

- Line 415: "In Figures 10 and 11, DSW outflow peaks in November and December in the Ross and Weddell seas, respectively" (and similar statements throughout this paragraph). My interpretation of Figures 10 and 11 is that they are showing the fraction of DSW, primarily on the shelf, and not the "outflow". To focus on the outflow would need an averaging area over the slope and a deeper depth range.

- Line 464: "The prominence of seasonality at the Antarctic margins will likely increase in the remainder of this century." I'm not sure this is true for DSW regions? If DSW stops being produced, I would say these regions will become LESS seasonal.

Regards,
Adele Morrison

References

Beadling, R. L. et al. Importance of the Antarctic Slope Current in the Southern Ocean Response to Ice Sheet Melt and Wind Stress Change. *J Geophys Res Oceans* 127, (2022).

Neme, J., England, M. H. & Hogg, A. McC. Projected Changes of Surface Winds Over the Antarctic Continental Margin. *Geophys Res Lett* 49, (2022).

Smith, R. S. et al. Coupling the U.K. Earth System Model to Dynamic Models of the Greenland and Antarctic Ice Sheets. *J. Adv. Model. Earth Syst.* 13, (2021).

Reviewer #2 (Remarks to the Author):

A review of "Seasonality of Intrusion of Warm Circumpolar Deep Water onto Antarctic Shelves" by Lanham et al. by Michael Haigh (BAS)

In this study the authors use a very high-resolution ocean model to examine the seasonal cycle of CDW transport onto the Antarctic continental shelf. The authors classify this transport into four different regimes. The results of this study are important since, as far as I'm aware, there is no other ice-ocean model in existence which can model these processes with such high resolution. One natural downside is the short timeframe (1 year) over which the analysis is conducted, but this is unavoidable given the computational demands of the model. For this reason, though, I think the authors need to be thorough in their physical reasoning for classifying the regimes for on-shelf CDW intrusions. My most major comment is that I don't think the authors are quite thorough or clear enough in this regard. After handling of this and the other major comments, I would be happy to recommend publication.

MAJOR COMMENTS

1. I think the paper would benefit from a little more description or clarification of the model SOHI. In particular, it's not clear what it means for the model to be "dynamically downscaled" from SOSE.

2. I'm unsure of the authors' reasoning for choosing the certain days/months when computing correlations. Why are daily

snapshots (or are they means?) used for one variable and monthly means used for the other? E.g., Fig. 5b shows correlations of on-shelf CDW taken on the 15th of each month with winds averaged over the month. I'm concerned that this statistic utilises winds that occur after the CDW sample, while the winds are being used to explain the CDW distributions.

3. I think two of the mechanisms driving on-shelf CDW intrusions need improved explanations.

Section 4.1.1. I think this section would benefit from an explicit mention of the processes that actually get CDW onto the continental shelf in this regime. The text says: "Antarctic easterlies across East Antarctica weaken...enabling enhanced CDW intrusion onto the shelf." In my view, the wind-driven coastal downwelling controls the export of CDW off of the continental shelf, so as the coastal easterlies weaken, less CDW is forced off of the shelf. I.e., the intrusions onto the shelf might not have changed. (Here I am equating the word "intrusion" with "heat fluxes across the shelf break".) Of course, the net effect is the same: more CDW on the shelf. But I think the distinction between the direction of the fluxes is important. I'd be interested to know if the authors disagree with the distinction I'm making.

I would also be interested to know the authors opinions on whether or not this coastal wind mechanism can be important in the undercurrent regime, and be a factor alongside the acceleration/deceleration of the undercurrent by shelf-break winds? If the authors think it could be a factor in the undercurrent regime, I think this is worth a mention in the paper.

Section 4.2.1 & L359 "Figure 8c shows.... This suggests that the seasonal export of DSW from the shelf enables the intrusion of CDW along the bottom of the shelf." When the DSW moves out of a particular area, it has to be replaced by something. Does this mean that the anticorrelations between CDW and DSW water mass volumes are guaranteed? More generally, I don't think the anticorrelation tells us the specific mechanism for CDW intrusion. What actually is the mechanism? Is it eddy transport of CDW generated as DSW flows down the continental slope, often characterised by the V-shaped density structure?

4. I found Figure 4 a little bit difficult to interpret. I suggest giving a short description in the caption of what the *1 and *2 mean, expanding on the legend provided. E.g., does *1 mean westerly wind anomalies specifically at the coast? For me the diagram for regime [2] (and to some extent [4]) isn't clear, and I couldn't understand this. I don't know what the arrows represent in regime [2].

MINOR COMMENTS

L34. "ocean-ice interface". Specify here that you mean the ocean-sea ice interface.

L40. Specify that you mean "intrusion" onto the continental shelf.

L42. By "buffer" do you mean "transport barrier"?

L47. Specify where the ASF is, i.e., at the above the continental slope.

L74-77. On these couple sentences here, I would say that we are actually not so certain that long-term changes are due to changing shelf-break winds. The future projections of Naughten et al. (2023, <https://doi.org/10.1038/s41558-023-01818-x>) show a warming shelf by an accelerated undercurrent, but doesn't find this to be correlated with the winds. I suggest mentioning there is this uncertainty in the mechanisms on long timescales. On the short timescales (e.g., monthly) that your study concerns, the evidence continues to suggest that winds control the undercurrent.

L109 & elsewhere. Use "1/24 degree" or "1/24°" rather than "1/24th of a degree".

L122. "eddy fluxes known to shape cross-slope CDW incursions". In some areas, e.g., Amundsen Sea, on-shelf transport is done by the time-mean flow.

L177 "below-300m agreement is markedly improved" Do you mean improved relative to above-300m?

207. "We assume...". In this sentence what is meant by "stronger constraint", and relative to what?

L227-232. I struggled to follow this paragraph, so I would suggest rewording to improve its clarity. E.g., I'm not sure what is being referred to in "This is typically on the 1st of each month".

L249. "CDW concentration at the sea floor". Can you justify using the seafloor a bit more. In areas where the CDW intrusions are linked to DSW flowing off of the shelf, can't the CDW lie above the DSW? Is using the seafloor always appropriate in this case?

L281, 282 & elsewhere. Inverted commas are the wrong way around.

Section 4.1.2. In this section, I suggest specifying which winds you're referring to. Am I right that here it's the shelf break winds, as opposed to the coastal winds in section 4.1.1?

Fig 7a,b. If the CDW fraction is averaged just over the first day of the month, and the undercurrent is averaged over the entire month, most of the undercurrent anomalies going into the computation of the average occur after the CDW data. Wouldn't we

need to look at CDW data that lags the undercurrent to have causality in the right direction?

Fig 7c. Is this correlation with bottom speed, rather than velocity? I wasn't sure what is meant by the word "translate". Is the claim that a faster undercurrent leads to faster bottom speeds on the shelf, or just that they are correlated? I think an extra sentence might be needed to explain why this link is important.

L403. "Figures 10c and 11c show the cross-correlation...". Don't these figures show annual timeseries, as stated in the figure captions?

L434. "a lack of easterly winds over West Antarctica prevents a substantial ASF development there". I may be wrong, but I believe the time-mean winds (as in ERA5) over the Amundsen and Bellingshausen Sea continental shelves are easterly.

L455. "mean time-ocean". Do you mean "time-mean ocean"?

Reviewer #3 (Remarks to the Author):

Manuscript COMMSENV-24-1867-T

QUESTIONS FROM THE JOURNAL

* What are the major claims of the paper?

The manuscript proposes "a unified view of CDW intrusion onto the Antarctic continental shelf" (line 11). It "advocate for a reformulation of the typical annual-mean regime classification of the Antarctic continental shelf" (lines 22-23).

* Are these claims novel, and will they be of interest to others in the community? If the conclusions are not original, it would be helpful if you could provide relevant references.

The "unified view of CDW intrusions" proposed by the authors is novel (as far as I'm aware). Its simplicity is attractive, and I can see it being of interest to others in the community.

This being said, the components involved in this "unified view" are not novel, and in this sense, the manuscript is closer to a synthesis/review than a research article. For example, the "undercurrent regime of seasonal CDW intrusion" (Section 4.1.2) will feel familiar to the readers of Dotto et al. 2019 (<https://doi.org/10.1175/JPO-D-19-0064.1>), Silvano et al. 2022 (<https://doi.org/10.1029/2022GL100646>)... The "DSW-induced CDW inflow along canyons" (Section 4.2.2) follows the work of Morrison et al. 2020 (<https://doi.org/10.1126/sciadv.aav2516>), Darelus et al. 2023 (<https://doi.org/10.1038/s41467-023-36580-3>)... The authors do acknowledge these earlier studies (as they should).

The authors' point that seasonality is critical to regime classifications of the Antarctic (especially "regime 3") is interesting and novel (as far as I'm aware).

* Is the work convincing?

The demonstrations are not always satisfying, in large part because it is challenging to present a huge synthesis (a "unified view") within the constraints of a regular research paper. Individual mechanisms such as the "undercurrent regime" typically deserve a full research paper (e.g. <https://doi.org/10.1175/JPO-D-19-0064.1>) but are here distilled in a few paragraphs (Section 4.1.2) which limits the discussion to qualitative aspects. Similarly, material such as Figs.8-11 are attractive to the eye and effective at qualitatively illustrating a mechanism, but 12 months of model outputs tagged with a R² statistics on top (panel (c) of Figs.8-11) is not a scientific demonstration that these mechanisms are at play in real life.

* What further evidence would be required to strengthen the conclusions?

The main document already includes 11 figures and a "Supplementary Material" of 17 pages and 14 supplementary figures, so I doubt the answer lies in requesting "more figures".

Instead, I would *revise the text* (Section 4) so that each of the 4 "regimes" (or "mechanisms") are firmly introduced in the context of the existing literature, before moving on to the model analyses. This would serve two purposes: (a) it would push the burden of demonstrating the existence of these 4 regimes/mechanisms to the literature, and (b) it would make it clearer to the reader that the "unified view"/synthesis is the novelty of the present manuscript. Note that the authors do cite the relevant literature here and there, but they do so as a secondary support to their model analyses. What I'm advising is to reverse the relationship---first ground the "4 regimes" in the literature, and only after that, use the model outputs as an illustration of the regimes in action. It would not be wise to attribute a larger role to the model outputs considering what they are---a limited and preliminary set of outputs from a new model (lines 152-156). The one year comparison with MEOP data (Fig.2) doesn't inform us about the model viability (long-term drifts, stability) or its skill after being allowed to run freely for 1-2 decades and thus form its own opinion of what the hydrography looks like.

* On a more subjective note, do you feel that the paper will influence thinking in the field in terms of either conceptual understanding or technological capability?

As I mentioned before, I think the "unified view of CDW intrusions" proposed by the authors is novel and attractive, and I can see it becoming useful to the community for conceptual understanding. The authors' conclusion that seasonality is critical to regime classifications of the Antarctic (lines 455-463) is valuable, and the implications of this conclusion for climate change (lines 464-475) are important for the community.

* Please feel free to raise any further questions and concerns about the paper.

I am concerned about the title ("Seasonality of intrusion..."). Such a title suggests a process-oriented study of the mechanisms behind the intrusions, which is not what the manuscript offers. I would recommend a title that better reflects the content: "A new classification of seasonal CDW intrusion onto Antarctic shelves", or something along those lines.

SPECIFIC COMMENTS

(1) Author affiliations: Affiliation #5 (Mashayek) is missing.

(2) Line 16: Acronym MEOP is nowhere defined.

(3) Lines 28-29: Symbols θ and S are used without having been defined.

(4) Line 40: The word "prevents" would imply that CDW does not exist at all in regions like the Ross Sea that produce DSW. But the historical data tell us that modified CDW does exist on the continental shelf of the Ross Sea. Perhaps "limits" instead of "prevents"?

(5) Lines 105, 167: I understand that SOSE is "robustly constrained by observational data" *in the deep, off-the-shelf portion* of the Southern Ocean where data are abundant. But when it comes to the continental shelf (the focus of this study), there are far fewer observational data points available (e.g. ARGO floats only rarely venture on the shelf). Unless the authors quantitatively and convincingly demonstrate that data assimilation taking place on the continental shelf is substantially improving SOSE's results on the continental shelf, I would modify this statement and acknowledge that the "robust constraints" are primarily improving the Southern Ocean, not so much the continental shelf. We don't want to mislead the readers of the journal.

(6) Line 109: I don't understand how the authors equate a $1/24^\circ$ resolution with "mesoscale/sub-mesoscale dynamics". In the meridional direction, $1/24^\circ$ is equivalent to 4.63km. How is a mesh size of 4.63km resolving the baroclinic Rossby radius of deformation on the weakly-stratified continental shelves (which are the topic of the study)? On the continental shelf of the Ross Sea, Mack et al. (<https://doi.org/10.1029/2018JC014688>) estimate this radius based on hydrography to be 1.7-2.5km (their Table 2). How is a $1/24^\circ$ resolution resolving that?

(7) Line 167: Circumpolar models are typically evaluated in terms of their hydrography (as done in Section 3.2 of the manuscript) but also their sea ice cover, and their basal ice shelf melting rates. How is SOHI doing for sea ice and ice shelf melting rates?

(8) Line 177: Is it 300m or 400m? The caption of Figure S2 says 400m, while line 177 says 300m.

(9) Line 251: Figure 5 is introduced before any mention of Figure 4.

(10) Caption of Figure 8: "The r-value of 0.97 is significant at the 99% level."

What does "significant" mean when you have 12 values for the blue (or red) curve, and each of the 12 values is highly correlated with the value before/after it (serial correlation)?

(11) Line 360: The text on line 360 reports " $r^2=0.93$ ". Fig.8c shows " $R^2=0.95$ ". The caption of Fig.8 mentions "r-value of 0.97". These inconsistencies are undermining my trust in the rigor of the statistical analyses. The reader expects you to: (a) define your symbols unambiguously, (b) use them consistently (lowercase *or* uppercase, not both; squared or not squared, not both), and (c) report the same value everywhere (don't change between the text, the figure, and the figure caption).

(12) Lines 485-695: The whole "References" section needs to be audited. The DOI is missing from multiple references, the publisher information is probably unnecessary for scientific articles and is often wrongly reported (e.g., line 489, 'Blackwell' isn't the publisher of JGR-Oceans), it's unclear to me that the journal wants a ISSN entry for scientific articles, some titles use sentence case while others use title case, some DOI links use the obsolete <http://dx.doi.org> in place of <https://doi.org>, information about volume is often missing for scientific articles... It appears that nobody (none of the coauthors, and nobody at the journal) bothered to review this section before it was sent to the reviewers.

(13) Supp.Materials, line 46: It is mentioned that "temperature and salinity converge to near 1 at ~35,000 data points" in Fig.S3f. However, Figs.S3a,c demonstrate how the number of blue points drops dramatically when requiring a large number of data points per month. How many blue points are left in the scatterplots when requiring 35,000 data points per month? Is the "r" value (Fig.S3f) obtained in such a case meaningful?

Communications Earth & Environment is committed to improving transparency in authorship. As part of our efforts in this direction, we are now requesting that all authors identified as 'corresponding author' create and link their Open Researcher and Contributor Identifier (ORCID) with their account on the Manuscript Tracking System prior to acceptance. ORCID helps the scientific community achieve unambiguous attribution of all scholarly contributions. You can create and link your ORCID from the home page of the Manuscript Tracking System by clicking on 'Modify my Springer Nature account' and following the instructions in the link below. Please also inform all co-authors that they can add their ORCIDs to their accounts and that they must do so prior to acceptance.

Version 1:

Decision Letter:

Dear Mr Lanham,

Your revised manuscript titled "Seasonal Regimes of Warm Circumpolar Deep Water Intrusion onto Antarctic Ice Shelves." has now been seen by our reviewers, whose comments appear below. In light of their advice we are delighted to say that we are happy, in principle, to publish a suitably revised version in Communications Earth & Environment, provided you include a robust interpretation of wind patterns over the Amundsen Sea, the role of barotropic wind forcing in driving the seasonal variability of the undercurrent, and a clear acknowledgement of the limitations of the statistical approach in the main text.

If you can address these requests, we therefore invite you to revise your paper one last time to address the remaining concerns of our reviewers. At the same time we ask that you edit your manuscript to comply with our format requirements and to maximise the accessibility and therefore the impact of your work.

EDITORIAL REQUESTS:

*****Please take care to match our formatting and policy requirements. We will check revised manuscript and return manuscripts that do not comply. Such requests will lead to delays. *****

SUBMISSION INFORMATION:

OPEN ACCESS:

Communications Earth & Environment is a fully open access journal. Articles are made freely accessible on publication. For further information about article processing charges, open access funding, and advice and support from Nature Research, please visit <https://www.nature.com/commsenv/open-access>

Link Redacted

Best regards,

Alireza Bahadori, PhD
Associate Editor
Communications Earth & Environment

REVIEWERS' COMMENTS:

Reviewer #1 (Remarks to the Author):

I am satisfied with the changes made by the authors and now recommend publication.

Reviewer #2 (Remarks to the Author):

Review of: "Seasonal Regimes of Warm Circumpolar Deep Water Intrusion onto Antarctic Ice Shelves"

I appreciate the authors' thorough response to each of my points in my original review, and I feel they have significantly improved the manuscript. In particular, I appreciate the transition from daily data to monthly data in some of the statistics, as this leads to a necessary improvement in the quality of these statistics. Overall, I find this manuscript continues to be a useful synthesis of pan-Antarctic mechanisms of cross-slope transfer, which makes good use of a state-of-the-art model. I therefore think that the manuscript deserves publication, but I do still have a handful of comments, which I think should be addressed before publication. While I have labelled some comments as major, as I think they are important, I also think they can be quickly handled by the authors.

Major

1. I still have an issue with discussion of winds over the Amundsen. In the rebuttal the authors says on-shelf Amundsen winds are not easterly, but they are. The near-zero time-mean winds to which the authors refer are over the shelf break, while it is the on-shelf winds that are responsible for the coastal downwelling.

"Holland et al. [28] show that greenhouse gas forcing has induced a transition from mean easterlies to near-zero mean winds in West Antarctica". The context of this sentence within the whole paragraph implies to me that you're referring to the on-shelf winds that are responsible for coastal downwelling. However, Holland et al. are referring to the winds in a very specific location over the shelf break in the Amundsen Sea. In the same paper they show there is no significant GHG-driven trend in the easterly winds over the continental shelf. Therefore, I don't think this result from Holland et al. is quite relevant for explaining the potential lack of ASF in West Antarctica.

2. In much of the discussion of the undercurrent regime, I think the authors are implying that a necessary feature of the undercurrent regime is a weak or non-existent ASF, which I don't think is accurate. First, I would say that there is a reasonably strong ASF in parts of the Amundsen Sea (see the observations of Walker et al. (2013)), and that here the ASF is responsible for the existence of the undercurrent in the first place. Importantly though, I don't think whether or not an ASF exists in this region is of great importance for the authors' conclusions (I understand it's unclear to what extent the ASF exists across West Antarctica). The authors' important conclusion is that seasonal variability in the undercurrent comes from barotropic wind-driven forcing. Whether or not there is a time-mean ASF or whether or not it is seasonally varying is a separate from this.

I suggest that the authors ensure they don't state or imply that a lack of ASF is a necessary feature for seasonal variability in cross-slope transfer to be driven by undercurrent variability. In particular, I would remove or edit the sentence at L393 ("The ASF is not..."), as this could imply that a lack of ASF is important for the undercurrent regime to exist. I make the same comment for the sentence at L528 ("a lack of strong...").

3. L320. Be careful with conflating winds and wind stresses. They are not necessarily oriented in the same direction, because of the sea ice. Can you clarify if here you are discussing the stress going into the ocean, through the combined effect of winds and sea ice? In Fig. 4 the title says "zonal wind", but caption says "wind stress". Which one is it? As I suggest above, if it's a stress, clarify as well if this is the actual stress going into the ocean, or some wind-only (ignoring sea ice) stress. I would be a cautious of drawing any conclusions if the latter is used, as sea ice is a critical factor in ocean surface stresses.

Minor

Title: "Seasonal Regimes of Warm Circumpolar Deep Water Intrusion onto Antarctic Ice Shelves". I wouldn't describe CDW as intruding "onto" ice shelves. Perhaps "towards".

I don't think abstract should have a section number. E.g., the introduction is section 1.

L52. 'ice shelf'  'ice shelves'

L54. 'continent'  'continental'

L98. "although it is also possible that, at longer timescales, buoyancy forcing changes may be dominant in warm regimes [32][33]." I'm not sure that Silvano et al. (2022, your ref. [32]) are talking about changes in buoyancy forcing in their paper, but rather baroclinic adjustments to variability in the wind stress curl (mechanical forcing). However, Haigh & Holland (2024) do show buoyancy forcing to be the driver of variability on longer (decadal) timescales in a warm shelf regime.

Fig 1. Briefly mention the depth of the dotted pink line. Is it possible to add shelf break contour to the temperature maps at the level of the pink line?

L163. Suggested changes: 'The model employs a Mercator projection...'. And 'The model has 225 vertical levels.'

L406. 'negative zonal flow anomalies'. State which direction (westward) rather than negative.

Fig 7. Does the correlation in Fig. 7c get stronger when the bottom flow speed is lagged behind the undercurrent speed? What are the advective timescales involved?

L433. "the Adelie shelf is sufficiently narrow to sustain a seasonal cycle of DSW formation and export". Could you add just a few extra words to this sentence? Is your reasoning that the Adelie shelf fills with DSW relatively quickly, maybe over a few months, so DSW export has to occur on these seasonal timescales?

Fig 10. It isn't clear in the upper panels: is the area shown wholly on the continental shelf?

References

Haigh, M., & Holland, P. R. (2024). Decadal variability of ice-shelf melting in the Amundsen Sea driven by sea-ice freshwater fluxes. *Geophysical Research Letters*, 51, e2024GL108406. <https://doi.org/10.1029/2024GL108406>

Walker, D. P., A. Jenkins, K. M. Assmann, D. R. Shoosmith, and M. A. Brandon (2013), Oceanographic observations at the shelf break of the Amundsen Sea, Antarctica, *J. Geophys. Res. Oceans*, 118, 2906–2918, doi:10.1002/jgrc.20212.

Reviewer #3 (Remarks to the Author):

I am concerned about the statistical analyses. The authors are limited by the model to a 1-year period (Line 168) which they analyze using monthly averages (Line 259). Based on e.g. Fig.8(c),9(c),10(c),11(c), the timeseries of model results are often dominated by a seasonal cycle. Quoting Emery & Thomson 2004:

"Since many data collected in time or space are highly correlated because of the presence of low-frequency, nearly deterministic components such as [...] the seasonal cycle, standard statistical methods do not really apply. Contrary to the requirements of stochastic theory, the values are not statistically independent. [...] A good example of this problem is presented by Chelton (1982) who showed that the high correlation between the integrated transport through Drake Passage in the Southern Ocean and the circumpolar-averaged zonal wind stress may largely be due to the presence of a strong semi-annual signal in both time series. A strong statistical correlation does not necessarily mean there is a cause and effect relationship between the variables."

So there is this first aspect that the correlations reported in the study may not be the causality that the authors are suggesting.

A second aspect has to do with statistical significance and the concept of degrees of freedom, which I believe the authors are misusing. For their 1-year long monthly timeseries (N=12), the authors "assume 10 degrees of freedom" (caption of Figs.4,6,7,8,9,10,11). But as stated above, a correlation requires the values within the timeseries to be statistically independent, which will not be the case for a sinusoid function of a 1 year period that has been sampled every month (i.e. a timeseries dominated by the seasonal cycle; Fig.8(c),9(c),10(c),11(c)). Emery & Thomson 2004 suggest in this case to compute the *effective* number of degrees of freedom that takes into account the presence of correlation between neighboring monthly values. The N-2=10 degrees of freedom assumed in the study is an overly optimistic estimate that assumes all points to be statistically independent.

Emery, W.J., R.E. Thomson, 2004, *Data analysis methods in physical oceanography*, Elsevier, 638 pages; quote from Section 5.2 "Stochastic processes and stationarity".

Response to Reviewers

We are grateful to the three Reviewers for their helpful and constructive feedback. In the following, we outline how we have responded to their comments. Comments by the Reviewers are shown in *blue*, and our responses in black. Line numbers refer to the revised version of the manuscript.

Response to Reviewer 1

The manuscript analyses one year of the SOHI model and identifies four regimes of seasonal CDW intrusion onto the Antarctic continental shelf. The watermass fraction analysis is a useful framework for untangling the complex dynamics of this region. The manuscript is a significant contribution to the field, and extends on the usual time-mean regime understanding of the Antarctic margins. The manuscript is well written with some very nice visualisations. I do however have some concerns, outlined further below, about the influence on the results of the large model drift and the choice of using a single day per month to diagnose seasonality. In addition, many regions in the correlation maps have low significance. Therefore I recommend major revisions. However, I did really enjoy reading this work and I look forward to reviewing a revised manuscript.

We thank the Reviewer for their extensive comments and advice on how to improve the manuscript. We have followed their recommendations, as detailed in the point-by-point responses below.

Major comments:

1. It would be helpful to add regions into the abstract for each of the 4 regimes, for those time-poor scientists who will skim the paper. e.g. regime 1 (East Antarctica), regime 2 (West Antarctica), regime 3 (Adelie coast), regime 4 (Ross/Weddell Seas).

Thank you - we have made this adjustment.

2. Line 84: "The response of shelf regimes to climate change is likely to hinge upon changes in the atmospheric circulation around Antarctica". Actually, I disagree with this statement. Yes, wind changes will undoubtedly play a role, but models projections (e.g. Beadling et al. 2022; Li et al. 2023) have shown that future change around Antarctica is likely to be dominated by meltwater forcing changes, not atmospheric forcing changes. In particular, regimes 3 and 4 that depend on DSW will likely be more dependent on changes in meltwater forcing. I suggest adding an additional paragraph after this one, focusing on the impact of potential meltwater forcing change, in addition to this paragraph on potential wind forcing change.

Good point. We have added in such a paragraph in lines 101-112.

3. An 8-month spinup period seems very short. Lines 160-165 hint that the drift in some regions is large (also as confirmed by Figures 8c and 9c, and even CDW in Figure 11c), but other regions are ok. The manuscript really needs to include validation to convince the readers that this short spinup is sufficient and that the seasonal variation that this paper focuses on is indeed seasonality and not drift. Options could be:

a) A plot showing a map of the magnitude of the drift in CDW fraction over the 1 year period, compared with the magnitude of the detrended seasonal cycle of CDW fraction at each location.

b) Time series from the start of the spinup to the end of the extended 2 month period after the 12 months analysed.

These plots would help the reader understand in which regions the results are likely to be robust, and which regions need to be considered more carefully.

If these plots show that the drift is larger than the seasonal cycle, then the output could be detrended before analysis.

Thank you for this comment, and also for those in [11] and [14]. We respond to all model-drift related points here. We recognise that we did not, originally, provide enough evaluation of the scale and impact of model drift.

It is initially worth noting that there is also a 12-month spin up at 1/12th degree resolution, plus the 8 months at 1/24th degree. This length is selected as it is sufficient to stabilise the kinetic energy in the model. Both of these additional pieces of information we now state in the main manuscript (lines 172 and 184 respectively).

However, it is true that there is evidence of possible T/S drift on some of the shelves. We now devote a section in the SI to model drift, which you can find at Section S4. We now have 18 months of model output (an additional 6 months from Sep_06-March_07), so we leverage this to investigate drift. But we summarise the main points here:

- There is evidence of a linear trend in water mass fractions in the Adelie Coast (as suggested by the Reviewer), but also in West Antarctica and an area of East Antarctica between 80-120E. This can be seen in Figure S14.
- However, in West Antarctica and 80-120E, there is also a significant linear trend in winds in the correct causality direction (i.e. in the same direction as the CDW trend). In these regions we suggest that it is hard to untangle forced trends from model drift, which is a natural disadvantage of having only 1 full year. This can be seen in Figure S14.
- In most regions, we retain statistical significance when we correlate the winds with the detrended seasonal cycle (see Figure S15). However, in some parts of the region 80-120E, correlations weaken (particularly the Knox coast is no longer significant). This is likely because (as mentioned above) there is a positive trend in both winds and CDW fraction over the year which contributes to the high r-values here.
- In the Adelie Coast, we extend the time series to 18-months in Figure S17a. This confirms the time series stabilises after ~1 year. We agree that the linear trend in the first 12 months could well be model drift.
- In order to rule out that the seasonal variance here is purely due to drift, we detrend the 18-month time series and calculate the seasonal variance again, shown in Figure S16. The Adelie Coast continues to show higher levels of seasonal variability than the other DSW-forming regions (Ross, Weddell, and Prydz Bay).
- Figure S17b shows the original time series figure, but for 18 months and de-trended. Figure S17c shows the same but for Prydz Bay (with the other 2 WMs). As expected, there is substantial variability in DSW formation in both regions. But in the Adelie Coast, this variability is balanced by CDW, and in Prydz Bay it is balanced by ISW (due to the differing shelf geometries). Our main conclusion is therefore unchanged.

We now make efforts to address this in the main manuscript. We acknowledge that there are linear trends in both CDW and wind in some parts of East Antarctica (regime 1) and West Antarctica (regime 2). We are more cautious about the timing of CDW maxima in these regions, stressing the focus is on the mechanism / relationship with the winds. Likewise, in the Adelie Coast, we acknowledge the possibility of drift, but make the point that the main conclusions remain the same regardless of detrending each time series. See lines 293-297, 365-367, 393-398, 425-435, and 448-450.

4. I understand the authors' reasoning behind using full depth MEOP data in the comparison in Figure 2, because there is more limited observational data available at depth. However, if possible, it would be a more appropriate and useful validation to only compare against MEOP in the deeper ocean (e.g. below 400m or where there is CDW), given that the focus of the paper is on CDW. I would be interested in seeing Figure S1 calculated only for a deeper subset (e.g. below 400m) of the MEOP data. I'm guessing that it may converge faster (i.e. a smaller number of data points may be acceptable), because there is likely to be less variation in T and S below the mixed layer. If this is the case, it would be a nice addition to add a second MEOP comparison figure into the main manuscript - e.g. add W. Ross and Knox T/S sub panels to Figure S2 and move it to Figure 2 (or even replace Figure 1?) in the main manuscript.

Thank you - this is a good point. We now show Figure S2 (originally S1) for below 400m in Figure S4 (for all regions). It appears as though convergence is somewhat quicker, but we still don't reach equilibrium in many regions. We do however include the other two most-sampled (>1500) regions (Knox, Ross) in Figure S3 (originally S2). Ross agreement remains good, but we lose a significant amount of agreement in the Knox Coast, which we now acknowledge in the main manuscript. It is unclear to us how much of this discrepancy is related to sampling issues. Figure S5f shows that there is a substantial risk of undersampling at this number of datapoints (< 5000) and we feel that it is probably preferable to keep the full depth comparison in the main manuscript.

It is worth noting that we now show SOHI-MEOP pairs in Figure S2f/S3e - which is why we have slightly fewer values (i.e. previously there were points near topography where the seals sample and this is sampled as topography in the model).

5. Line 227: I really like this water mass fraction analysis! However, it seems to me that it would be better to use monthly averages for the water mass analysis, rather than a single daily average each month. Monthly averages would be more representative of the whole month. Especially for the watermass variance shown in Figure 3, the high variance here could just be the result of eddies in particular daily averages rather than high seasonal variance. Also for Figure 5b, the correlation is between the `_monthly_` wind stress and the `_daily_` watermass fraction. It would make much more sense here to use the same time period of averaging for both quantities before correlating. I don't understand why using daily averages would be computationally easier than using monthly averages. Once monthly averages are computed, then there would have the same number of data points as using a single day each month.

Thank you, we have updated these plots to now reflect monthly averages. All plots now show monthly mean fields unless otherwise stated.

6. Line 243: "The lack of cavities means that SOSE does not represent significant areas of DSW production in the Ross and Weddell seas." I disagree with this logic. Other models (e.g. ACCESS-OM2-01, Moorman et al. 2020) produce DSW without ice shelf cavities. You can also see in Figure 3c that the DSW fraction is highest over the open continental shelf regions where winds and atmospheric cooling are high, not under cavities (except in the western Ross, where I'm guessing the DSW forms on the open continental shelf and flows under the cavity).

Agreed, this was an oversight. We have now changed this sentence (line 273).

7. It would be a useful addition to add a map similar to Figure 5, but with all locations with high CDW fractions ($>0.3?$) shown (not just those with high variance). There are areas with strong CDW intrusions (e.g. Drygalski trough, Amery) which don't show up in Figure 5. Is the seasonality in these regions really close to zero? I'm wondering if the current variance metric could be dominated by high frequency (eddy) variations in some locations, e.g. variance is high along the shelf break around much of East Antarctica, and the peak months are quite spatially inhomogeneous in many regions (east of Amery, George VI). Such an additional map figure could go in the supplementary if needed. If the true seasonal variance in these other locations is indeed weak, then this is not needed.

As stated in the response to point [5], we have now updated this analysis to include monthly-mean fields.

8. There are many areas of the correlations shown in Figure 5b that are not significant to the 90% level. I suggest only showing areas on Figure 5 that are significant, or adding stippling as done in Figure 7c. If non-significant regions are masked, then the current map could be added to Figure S6 instead.

Thank you, we have updated the Figure to include stippling.

9. I really like the regime classification in Figure 4. However, I think it would be easier to interpret (and more likely to be reused in other people's talks etc) if the schematics were enlarged and had a little more detail. e.g. For the ASF regime it would be useful to show strong winds for the top panel and weak winds for the lower panel. I struggle to interpret what the undercurrent regime schematic is showing - CDW moving upwards in the water column and travelling westward along the shelf break? How is that relevant for intrusions onto the shelf? Is the lower line the undercurrent? But it's travelling westward not eastward? If it is the undercurrent, it could be labelled. For the Canyon Regime, is the upper line SSH? Or an isopycnal? This could be labelled.

Thank you. We have updated the Figure to include your suggestions, including a reformation of the regime 2 schematic.

10. Caption of Figure 6: "These correlations are significant at the 99% level over 20-60°E and at the 90% level over 80-130°E (see Figure S6)." This does not seem consistent with what is shown in Figure S6. From Figure S6c, it looks like most of 20-60°E is closer to 95% significance, with only limited points in that longitude range being significant at 99%. Between 80-130°E, it looks to me like most points are not even significant at 90% (Figure S6a). As per

my comment above for Figure 5, I suggest limiting what is shown in Figure 6 to only include points that are deemed significant; or add stippling.

Thank you, we have added stippling in the relevant Figures and updated their captions accordingly.

11. Lines 359-362: "Figure 8c shows that the seasonal cycle of CDW concentration at the bottom of the Adelie shelf is strongly anticorrelated with that of DSW ($r^2 = 0.93$). This suggests that the seasonal export of DSW from the shelf enables the intrusion of CDW along the bottom of the shelf." I strongly disagree with this conclusion. As the authors acknowledge as a possibility later in this paragraph, I think Figure 8c (also 9c) predominantly shows model drift, not seasonality. It looks to me like the model is not producing as much DSW compared to what is present in the initial conditions, so that the fraction of DSW is tending downwards (both in Adelie and Prydz). As the DSW fraction tends downwards, this watermass is instead being replaced by CDW. Figure S10 shows that there is some seasonal production of DSW, but the annual production magnitude looks to be much smaller than the model drift. I find it problematic that the reason the Adelie coast is showing up with high variance in Figure 3b,d is likely just due to model drift and not seasonal variability. As suggested above, perhaps this could be addressed by detrending each location first before calculating the seasonality/variance shown in Figures 3 and 5.

Line 369: "It is possible that this could result from model drift". This seems like the most logical conclusion to draw. It would be useful to increase the length of the time series in Figure S10 to include the first 8 months of the spinup to clarify this.

Line 372-373: "Regardless, the strength of the relationship between DSW and CDW on this shelf is of primary mechanistic importance." I disagree that you can conclude anything about mechanisms from Figure 8c if you cannot rule out the possibility that it is just drift.

Thank you. For simplicity, we respond to all model drift related points (i.e. points 3, 11, and 14) in the response to point 3.

12. Line 378: "even when DSW is exported". I have not seen any evidence in the rest of the paper that DSW is exported to the abyss in this model. From Figure 3c it looks to me like there is no connection between the DSW watermass on the shelf and in the abyss (e.g. in the Ross and Weddell). Instead there is a ring of CDW at the ocean floor at the base of the continental slope. I would have thought that even after 1 year, you might see overflows reaching deeper down the slope if they are present in the model. To make a statement like this would require showing that DSW is exported off the shelf (to the abyss or mid depths?) rather than just mixed away.

Agreed – we have updated the language (line 440).

13. Lines 387-388: "the seasonal cycle in DSW formation, which reaches a maximum in winter". I don't see this in Figure 9c? To me it looks like the DSW fraction is minimum in winter (June/July)?

Thank you for bringing this to our attention. The line in question has been removed.

14. I like the time series in Figures 8-11, because they help show the magnitude of CDW seasonal variability, and put this into the context of model drift. It would be nice to add similar time series (perhaps with LSSW instead of DSW) to Figures 6 and 7 for the East and West Antarctic regions.

As above: we respond to all model drift related points in the response to point 3.

Minor comments

- Line 28: "affected" not "effected"?

Sentence has now been updated, thanks (line 29).

- Paragraph at line 84: It would be helpful to also cite Neme et al. 2022 here, which analyses future projections of change in the easterly winds close to Antarctica.

Thank you, done.

- Line 123: "GCMs also do not include ice shelf cavities." This is a bit of a sweeping generalisation. e.g. What about UKESM (Smith et al. 2021)? Suggest adding "The majority of..." or similar.

Updated as suggested (line 142).

- Line 135: Is sea ice forcing included in the atmospheric forcing that's applied? Best to specify this here.

There is a coupled sea ice model and both the sea ice and ocean models are driven by ERA5. We now note this explicitly in the text (lines 155-156).

- Figure 2: The outside subpanels need 'Temperature' and 'Salinity' labels on the axes.

Thank you, done.

- Caption of Figure 2: "Due to computational constraints, these plots include just the first 6 months of SOI data (September - March)". I am confused by this statement. Does this mean that no MEOP data from the months April - October is included in the comparison, because it is not possible to sample the model in those months? I also don't understand why this is computationally infeasible.

We have now adapted to the plot to include all months.

- I suggest reducing the use of acronyms. Acronyms reduce the readability of the paper. In particular, I don't think GCM, SWT and WMT are needed.

Thanks, suggested acronyms are removed.

- Figure S4 is not referenced in the main text.

Now referenced (Line 212).

- Figure 3a - Is this SOSE or SOHI? It would help to add the model to the title as is done for panels c and e. Or else remove the SOHI label from panel c if all other panels except e refer to SOHI.

Thank you, updated.

- The ordering of the Supplementary figures does not match the order they are called in the main manuscript. e.g. Figure S7 is called before Figure S6.

The order of the Supplementary figures matched the order in which they are called in the Supplementary Materials document, but we have now changed this to reflect the order in which they are called in the main manuscript .

- Figure 5 is discussed (line 251) before Figure 4 is discussed (line 280). I suggest swapping the order of these figures.

Thank you, done.

- The placement of the colorbar in Figure 5a covers the longitude labels and is too close to panel b.

Thank you, updated.

- Line 402: "green" should be "blue".

Thank you, updated (line 466).

- Line 413-414: "Surface WMT is initiated in March in the Ross Sea, and in April in the Weddell Sea" (and similar statements throughout this paragraph). Does this refer to the seasonality from Schmidt et al. 2023 or SOHI? I would expect this to be model dependent, because the calculation depends on model surface density (and therefore model advection timescales etc). Therefore I don't think it's a relevant comparison to use the timing directly from Schmidt et al. 2023 if that is what is done here. However this is straightforward calculation to redo from the SOHI model if required, only requiring surface fluxes/density.

Good point. We have decided to remove this discussion from the manuscript.

- Line 415: "In Figures 10 and 11, DSW outflow peaks in November and December in the Ross and Weddell seas, respectively" (and similar statements throughout this paragraph). My interpretation of Figures 10 and 11 is that they are showing the fraction of DSW, primarily on the shelf, and not the "outflow". To focus on the outflow would need an averaging area over the slope and a deeper depth range.

Agreed that the averaging area does not necessarily describe outflow. As we state in the response to the previous point, we have decided to remove this discussion from the manuscript.

- Line 464: "The prominence of seasonality at the Antarctic margins will likely increase in the remainder of this century." I'm not sure this is true for DSW regions? If DSW stops being produced, I would say these regions will become LESS seasonal.

Thank you for raising this. We have updated the language in this sentence to say that 'some regions' 'may' experience increased seasonality (line 514).

Response to Reviewer 2

In this study the authors use a very high-resolution ocean model to examine the seasonal cycle of CDW transport onto the Antarctic continental shelf. The authors classify this transport into four different regimes. The results of this study are important since, as far as I'm aware, there is no other ice-ocean model in existence which can model these processes with such high resolution. One natural downside is the short timeframe (1 year) over which the analysis is conducted, but this is unavoidable given the computational demands of the model. For this reason, though, I think the authors need to be thorough in their physical reasoning for classifying the regimes for on-shelf CDW intrusions. My most major comment is that I don't think the authors are quite thorough or clear enough in this regard. After handling of this and the other major comments, I would be happy to recommend publication.

We thank the Reviewer for their comments and suggestions on how to improve the manuscript, which we address below.

Major Comments

1. I think the paper would benefit from a little more description or clarification of the model SOHI. In particular, it's not clear what it means for the model to be "dynamically downscaled" from SOSE.

Thank you. We have added a few more sentences in Section 3.1 describing the model. We have also changed 'dynamically downscaled' to 'initialised' to improve clarity (line 160). We would like to emphasise again that a thorough introduction to the model (+ validation) can be found in Dinh et al. 2024, as we reference in the paper.

2. I'm unsure of the authors' reasoning for choosing the certain days/months when computing correlations. Why are daily snapshots (or are they means?) used for one variable and monthly means used for the other? E.g., Fig. 5b shows correlations of on-shelf CDW taken on the 15th of each month with winds averaged over the month. I'm concerned that this statistic utilises winds that occur after the CDW sample, while the winds are being used to explain the CDW distributions.

Thank you for raising this. There were initially no monthly diagnostics saved. We have now computed them, and have updated the majority of Figures with a monthly mean instead. Specifically to the point raised here, all wind correlations now use a monthly mean in both

wind and water mass fractions. There are a few cases in which daily averages are used, and these are now stated explicitly.

3. I think two of the mechanisms driving on-shelf CDW intrusions need improved explanations.

Section 4.1.1. I think this section would benefit from an explicit mention of the processes that actually get CDW onto the continental shelf in this regime. The text says: "Antarctic easterlies across East Antarctica weaken...enabling enhanced CDW intrusion onto the shelf." In my view, the wind-driven coastal downwelling controls the export of CDW off of the continental shelf, so as the coastal easterlies weaken, less CDW is forced off of the shelf. I.e., the intrusions onto the shelf might not have changed. (Here I am equating the word "intrusion" with "heat fluxes across the shelf break".) Of course, the net effect is the same: more CDW on the shelf. But I think the distinction between the direction of the fluxes is important. I'd be interested to know if the authors disagree with the distinction I'm making.

I would also be interested to know the authors opinions on whether or not this coastal wind mechanism can be important in the undercurrent regime, and be a factor alongside the acceleration/deceleration of the undercurrent by shelf-break winds? If the authors think it could be a factor in the undercurrent regime, I think this is worth a mention in the paper.

Section 4.2.1 & L359 "Figure 8c shows.... This suggests that the seasonal export of DSW from the shelf enables the intrusion of CDW along the bottom of the shelf." When the DSW moves out of a particular area, it has to be replaced by something. Does this mean that the anticorrelations between CDW and DSW water mass volumes are guaranteed? More generally, I don't think the anticorrelation tells us the specific mechanism for CDW intrusion. What actually is the mechanism? Is it eddy transport of CDW generated as DSW flows down the continental slope, often characterised by the V-shaped density structure?

Response to point a:

Thank you. We agree that wind-driven coastal downwelling removes CDW from the shelf when easterlies are strong, and have now acknowledged this process explicitly in the main text. However, in our view, this is not the only relevant process: when the isopycnals flatten, the along-isopycnal mixing of CDW/heat is much more efficient than when the ASF is fully formed (and so mixing must be diapycnal). So we would probably tend to disagree with the second part of your statement that the heat fluxes across the shelf break might not have changed. Regardless, for the sake of simplicity, we now refer to CDW intrusions in this regime as 'net on-shelf transfer of CDW', as it seems that the net effect is most important here (line 362).

Response to point b:

Thank you, this is a good point. We agree that the reduced coastal downwelling from westerly winds could also be a contributing factor in the CDW-wind correlations that we observe here. We have now added this to the main manuscript (lines 380-381).

Response to point c:

'Does this mean that the anticorrelations between CDW and DSW water mass volumes are guaranteed?': The short answer is no, but the water mass fractions must add up to (very nearly) 1. So if DSW is exported from the shelf, something must take its place. In this case, CDW fractions increase in an almost equal and opposite manner to DSW. But it could equally be low salinity shelf water (LSSW) or ice shelf water / winter water (which we label ISW). In the contrasting example we give, in Prydz Bay, the seasonal export of DSW doesn't bring the same level of CDW intrusion (it is replaced by LSSW/ISW instead) due to contrasting shelf geometry. So in this case the correlation breaks down and is no longer significant (0.59 in Prydz vs 0.97 in Adelie). So the point we make here is that the Adelie Coast is the only DSW-producing region which shows seasonal variability in CDW fractions.

'What actually is the mechanism?': This is an important point, thank you. We suggest that, fundamentally, it is likely to be a circulation set up by mass conservation, as suggested by Snow et al. (2016) in this region i.e. the buoyancy driven off-shelf flow of DSW is balanced by on-shelf flow of CDW. We have now added an extra sentence in the hope of clarifying this in the main manuscript. There are a number of mechanisms that could mediate this return flow. As you say, eddies could play a role, as could bathymetrically-steered flows. Whilst we feel that this is probably somewhat out of the scope of this (already-long!) paper, we do now acknowledge this in the main discussion (lines 420-422).

4. I found Figure 4 a little bit difficult to interpret. I suggest giving a short description in the caption of what the *1 and *2 mean, expanding on the legend provided. E.g., does *1 mean westerly wind anomalies specifically at the coast? For me the diagram for regime [2] (and to some extent [4]) isn't clear, and I couldn't understand this. I don't know what the arrows represent in regime [2].

Thank you. We have updated regime 2 in the schematic and have expanded the description of *1 and *2 in the figure caption.

Minor Comments

L34. "ocean-ice interface". Specify here that you mean the ocean-sea ice interface.

Thank you, done (line 35).

L40. Specify that you mean "intrusion" onto the continental shelf.

Updated (line 41).

L42. By "buffer" do you mean "transport barrier"?

Yes, thank you. Updated (line 44).

L47. Specify where the ASF is, i.e., at the above the continental slope.

Done, thank you (line 49).

L74-77. On these couple sentences here, I would say that we are actually not so certain that long-term changes are due to changing shelf-break winds. The future projections of Naughten et al. (2023, <https://doi.org/10.1038/s41558-023-01818-x>) show a warming shelf by an accelerated undercurrent, but doesn't find this to be correlated with the winds. I suggest mentioning there is this uncertainty in the mechanisms on long timescales. On the short timescales (e.g., monthly) that your study concerns, the evidence continues to suggest that winds control the undercurrent.

Good point - thank you. We have updated the language in the lines in question (lines 82-83), and have also added another paragraph in the introduction discussing the role of buoyancy forcing in future change around the Antarctic margin (lines 101-112).

L109 & elsewhere. Use "1/24 degree" or "1/24°" rather than "1/24th of a degree".

Thank you, updated.

L122. "eddy fluxes known to shape cross-slope CDW incursions". In some areas, e.g., Amundsen Sea, on-shelf transport is done by the time-mean flow.

Agreed. Updated now to '*some of the cross-slope..*'. (lines 141-142)

L177 "below-300m agreement is markedly improved" Do you mean improved relative to above-300m?

Currently fig 2. includes observations from all depths where we have MEOP data. By this we mean that excluding the top 400m improves the agreement. So it is improved relative to the full depth range. We now specify this in line 204.

207. "We assume...". In this sentence what is meant by "stronger constraint", and relative to what?

This is discussed in detail in the Supplementary Information in Section S2.3. But we force the Least Squares solution to adhere more strongly to returning a residual of 0 in the mass conservation equation. A stronger constraint means it has a large weight in the cost function - we place a larger weight on this condition.

L227-232. I struggled to follow this paragraph, so I would suggest rewording to improve its clarity. E.g., I'm not sure what is being referred to in "This is typically on the 1st of each month".

Thank you, this has now been reworded.

L249. "CDW concentration at the sea floor". Can you justify using the seafloor a bit more. In areas where the CDW intrusions are linked to DSW flowing off of the shelf, can't the CDW lie above the DSW? Is using the seafloor always appropriate in this case?

This is a good point and is certainly true in regions of DSW overflow (elsewhere we can be reasonably confident that CDW is the densest class of water mass). However, given the

resolution of the model and the memory required to perform even basic circumpolar computations, we need to define some depth level to make it manageable computationally. The seafloor is the only 'depth level' in which we can be sure of capturing all horizontal locations, particularly in the cavity. Moreover, looking at the sample water mass classification output in Figure S7 - it looks as though all CDW intrusions onto the shelf are concentrated at the sea floor (even in the Weddell and Ross Seas where DSW is formed). That being said, we have now added text to say that we acknowledge that we may miss some CDW intrusions may occur above the sea floor, as this is not an exhaustive analysis (lines 281-287).

L281, 282 & elsewhere. Inverted commas are the wrong way around.

Thank you, updated.

Section 4.1.2. In this section, I suggest specifying which winds you're referring to. Am I right that here it's the shelf break winds, as opposed to the coastal winds in section 4.1.1?

Yes, thank you. We have specified this in lines 380, 381 and elsewhere.

Fig 7a,b. If the CDW fraction is averaged just over the first day of the month, and the undercurrent is averaged over the entire month, most of the undercurrent anomalies going into the computation of the average occur after the CDW data. Wouldn't we need to look at CDW data that lags the undercurrent to have causality in the right direction?

Thank you. As stated on a reply to a previous point, these now use monthly-mean values.

Fig 7c. Is this correlation with bottom speed, rather than velocity? I wasn't sure what is meant by the word "translate". Is the claim that a faster undercurrent leads to faster bottom speeds on the shelf, or just that they are correlated? I think an extra sentence might be needed to explain why this link is important.

Thank you for raising this - we do in fact mean bottom speed. We have updated the text and figure to reflect this, and revised the language to improve clarity (lines 400-404).

L403. "Figures 10c and 11c show the cross-correlation...". Don't these figures show annual timeseries, as stated in the figure captions?

Yes, thank you. Updated (line 467).

L434. "a lack of easterly winds over West Antarctica prevents a substantial ASF development there". I may be wrong, but I believe the time-mean winds (as in ERA5) over the Amundsen and Bellingshausen Sea continental shelves are easterly.

It is our understanding that the zonal winds here are near-zero in the mean. See, for example, Holland et al. (2019) 'West Antarctic ice loss influenced by climate variability and anthropogenic forcing'. We have also updated the sentence to 'a lack of *strong* easterly winds' (line 484).

L455. "mean time-ocean". Do you mean "time-mean ocean"?

Yes, thank you. Updated (line 505).

Response to Reviewer 3

Questions from the Journal

We thank the Reviewer for their comments and thoughts on how to improve the manuscript. Our responses are outlined below.

* What are the major claims of the paper?

The manuscript proposes "a unified view of CDW intrusion onto the Antarctic continental shelf" (line 11). It "advocate for a reformulation of the typical annual-mean regime classification of the Antarctic continental shelf" (lines 22-23).

* Are these claims novel, and will they be of interest to others in the community? If the conclusions are not original, it would be helpful if you could provide relevant references.

The "unified view of CDW intrusions" proposed by the authors is novel (as far as I'm aware). Its simplicity is attractive, and I can see it being of interest to others in the community. This being said, the components involved in this "unified view" are not novel, and in this sense, the manuscript is closer to a synthesis/review than a research article. For example, the "undercurrent regime of seasonal CDW intrusion" (Section 4.1.2) will feel familiar to the readers of Dotto et al. 2019 (<https://doi.org/10.1175/JPO-D-19-0064.1>), Silvano et al. 2022 (<https://doi.org/10.1029/2022GL100646>)... The "DSW-induced CDW inflow along canyons" (Section 4.2.2) follows the work of Morrison et al. 2020 (<https://doi.org/10.1126/sciadv.aav2516>), Darelus et al. 2023 (<https://doi.org/10.1038/s41467-023-36580-3>)... The authors do acknowledge these earlier studies (as they should).

The authors' point that seasonality is critical to regime classifications of the Antarctic (especially "regime 3") is interesting and novel (as far as I'm aware).

* Is the work convincing?

The demonstrations are not always satisfying, in large part because it is challenging to present a huge synthesis (a "unified view") within the constraints of a regular research paper. Individual mechanisms such as the "undercurrent regime" typically deserve a full research paper (e.g. <https://doi.org/10.1175/JPO-D-19-0064.1>) but are here distilled in a few paragraphs (Section 4.1.2) which limits the discussion to qualitative aspects. Similarly, material such as Figs.8-11 are attractive to the eye and effective at qualitatively illustrating a mechanism, but 12 months of model outputs tagged with a R² statistics on top (panel (c) of Figs.8-11) is not a scientific demonstration that these mechanisms are at play in real life.

Thank you. For convenience, we address both this and the following comment jointly (see below).

* What further evidence would be required to strengthen the conclusions?

The main document already includes 11 figures and a "Supplementary Material" of 17 pages and 14 supplementary figures, so I doubt the answer lies in requesting "more figures". Instead, I would *revise the text* (Section 4) so that each of the 4 "regimes" (or "mechanisms") are firmly introduced in the context of the existing literature, before moving on to the model analyses. This would serve two purposes: (a) it would push the burden of demonstrating the existence of these 4 regimes/mechanisms to the literature, and (b) it would make it clearer to the reader that the "unified view"/synthesis is the novelty of the present manuscript. Note that the authors do cite the relevant literature here and there, but they do so as a secondary support to their model analyses. What I'm advising is to reverse the relationship---first ground the "4 regimes" in the literature, and only after that, use the model outputs as an illustration of the regimes in action. It would not be wise to attribute a larger role to the model outputs considering what they are---a limited and preliminary set of outputs from a new model (lines 152-156). The one year comparison with MEOP data (Fig.2) doesn't inform us about the model viability (long-term drifts, stability) or its skill after being allowed to run freely for 1-2 decades and thus form its own opinion of what the hydrography looks like.

Thank you. We have now made efforts to stress that what we offer is a unified view of CDW intrusion on the continental shelf. We agree that this is the novelty of this study. This unified view integrates results from regional studies - some of which provide more detailed mechanistic insights - and presents them in a new way which sheds light on why mechanisms change from region to region. We are now more explicit about this at the start - please see the final paragraph of the Introduction (lines 124-131). We agree that the restructuring of the paper, as suggested above, could provide a nice alternative presentation. But given the original structure of the paper seemed appealing to the other two reviewers, we chose not to restructure the paper in a major way.

* On a more subjective note, do you feel that the paper will influence thinking in the field in terms of either conceptual understanding or technological capability?

As I mentioned before, I think the "unified view of CDW intrusions" proposed by the authors is novel and attractive, and I can see it becoming useful to the community for conceptual understanding. The authors' conclusion that seasonality is critical to regime classifications of the Antarctic (lines 455-463) is valuable, and the implications of this conclusion for climate change (lines 464-475) are important for the community.

* Please feel free to raise any further questions and concerns about the paper.

I am concerned about the title ("Seasonality of intrusion..."). Such a title suggests a process-oriented study of the mechanisms behind the intrusions, which is not what the manuscript offers. I would recommend a title that better reflects the content: "A new classification of seasonal CDW intrusion onto Antarctic shelves", or something along those lines.

We have now adopted the following title: Seasonal Regimes of Warm CDW Intrusion onto Antarctic Ice Shelves.

Specific comments

(1) Author affiliations: Affiliation #5 (Mashayek) is missing.

Thank you, updated.

(2) Line 16: Acronym MEOP is nowhere defined.

Thank you for pointing this out, we have now defined it in line 195.

(3) Lines 28-29: Symbols θ and S are used without having been defined.

Thank you. Updated (lines 29-30).

(4) Line 40: The word "prevents" would imply that CDW does not exist at all in regions like the Ross Sea that produce DSW. But the historical data tell us that modified CDW does exist on the continental shelf of the Ross Sea. Perhaps "limits" instead of "prevents"?

Agreed, thank you. Updated (line 41).

(5) Lines 105, 167: I understand that SOSE is "robustly constrained by observational data" *in the deep, off-the-shelf portion* of the Southern Ocean where data are abundant. But when it comes to the continental shelf (the focus of this study), there are far fewer observational data points available (e.g. ARGO floats only rarely venture on the shelf). Unless the authors quantitatively and convincingly demonstrate that data assimilation taking place on the continental shelf is substantially improving SOSE's results on the continental shelf, I would modify this statement and acknowledge that the "robust constraints" are primarily improving the Southern Ocean, not so much the continental shelf. We don't want to mislead the readers of the journal.

This is true, thank you. We have softened the language and acknowledged this explicitly in Section 3.1 (lines 161-163).

(6) Line 109: I don't understand how the authors equate a 1/24deg resolution with "mesoscale/sub-mesoscale dynamics". In the meridional direction, 1/24deg. is equivalent to 4.63km. How is a mesh size of 4.63km resolving the baroclinic Rossby radius of deformation on the weakly-stratified continental shelves (which are the topic of the study)? On the continental shelf of the Ross Sea, Mack et al. (<https://doi.org/10.1029/2018JC014688>) estimate this radius based on hydrography to be 1.7-2.5km (their Table 2). How is a 1/24deg resolution resolving that?

It is a Mercator projection so near the shelf 1/24th is ~1.2km at 75S (in both dx and dy). We have now added this to the revised manuscript (lines 149-150).

(7) Line 167: Circumpolar models are typically evaluated in terms of their hydrography (as

done in Section 3.2 of the manuscript) but also their sea ice cover, and their basal ice shelf melting rates. How is SOHI doing for sea ice and ice shelf melting rates?

Good question. Analysis of SOHI's pan-Antarctic sea ice cover is done in Dinh et al. 2024 (<https://agupubs.onlinelibrary.wiley.com/doi/full/10.1029/2023GL106377>). To briefly summarise: they compare SOHI with SOSE, ECCO4, LLC4320, and satellite obs. All models show a 10-20% sea ice overestimation in winter and a shortened melt season. They also do some evaluation of melt rates in their Discussion. Given the current significant length of the manuscript and SI, we feel that further analysis of this nature is probably out of the scope of this paper. However, we have added an extra sentence in Section 3.2 to point the reader to Dinh et al. (2024) for further validation (lines 215-216).

(8) Line 177: Is it 300m or 400m? The caption of Figure S2 says 400m, while line 177 says 300m.

It is 400m. Thank you for bringing this to our attention, it has now been rectified.

(9) Line 251: Figure 5 is introduced before any mention of Figure 4.

Thank you, we have changed the order.

(10) Caption of Figure 8: "The r-value of 0.97 is significant at the 99% level." What does "significant" mean when you have 12 values for the blue (or red) curve, and each of the 12 values is highly correlated with the value before/after it (serial correlation)?

The r-value is 0.97 and is significant at the 99% level assuming we have 10 degrees of freedom (N-2). Of course, this assumes that the points are independent. If we assume that this is not the case, i.e. the data are not independent, we can re-test the significance assuming half the effective degrees of freedom. In this case, the p-value of the correlation is 0.001 - so it remains significant at the 99% level. In any case, we would argue that we have more degrees of freedom than this. To aid clarity, we now state how many degrees of freedom we assume in the relevant Figure captions.

(11) Line 360: The text on line 360 reports " $r^2=0.93$ ". Fig.8c shows " $R^2=0.95$ ". The caption of Fig.8 mentions "r-value of 0.97". These inconsistencies are undermining my trust in the rigor of the statistical analyses. The reader expects you to: (a) define your symbols unambiguously, (b) use them consistently (lowercase *or* uppercase, not both; squared or not squared, not both), and (c) report the same value everywhere (don't change between the text, the figure, and the figure caption).

Thank you, we have now defined all r-values as lowercase and not squared.

(12) Lines 485-695: The whole "References" section needs to be audited. The DOI is missing from multiple references, the publisher information is probably unnecessary for scientific articles and is often wrongly reported (e.g., line 489, 'Blackwell' isn't the publisher of JGR-Oceans), it's unclear to me that the journal wants a ISSN entry for scientific articles, some titles use sentence case while others use title case, some DOI links use the obsolete <http://dx.doi.org> in place of <https://doi.org>, information about volume is often missing for

scientific articles... It appears that nobody (none of the coauthors, and nobody at the journal) bothered to review this section before it was sent to the reviewers.

Thank you, we have fixed those references that were cited somewhat sloppily.

(13) Supp.Materials, line 46: It is mentioned that "temperature and salinity converge to near 1 at ~35,000 data points" in Fig.S3f. However, Figs.S3a,c demonstrate how the number of blue points drops dramatically when requiring a large number of data points per month. How many blue points are left in the scatterplots when requiring 35,000 data points per month? Is the "r" value (Fig.S3f) obtained in such a case meaningful?

Good point. At > 35,000, there are 8 blue points. The r-value at this level for salinity is 0.98 and theta is 0.99. Assuming 6 effective degrees of freedom, these values are easily significant at the 99% level ($p=0.00002$ and 0.00001 for S and T, respectively) .

Response to Reviewers, Version 2

Once more, we are grateful to the three Reviewers for their helpful and constructive feedback. In the following, we outline how we have responded to their comments. Comments by the Reviewers are shown in *blue*, and our responses in black. Line numbers refer to the revised version of the manuscript.

Response to Reviewer 2

I appreciate the authors' thorough response to each of my points in my original review, and I feel they have significantly improved the manuscript. In particular, I appreciate the transition from daily data to monthly data in some of the statistics, as this leads to a necessary improvement in the quality of these statistics. Overall, I find this manuscript continues to be a useful synthesis of pan-Antarctic mechanisms of cross-slope transfer, which makes good use of a state-of-the-art model. I therefore think that the manuscript deserves publication, but I do still have a handful of comments, which I think should be addressed before publication. While I have labelled some comments as major, as I think they are important, I also think they can be quickly handled by the authors.

Major

1. I still have an issue with discussion of winds over the Amundsen. In the rebuttal the authors says on-shelf Amundsen winds are not easterly, but they are. The near-zero time-mean winds to which the authors refer are over the shelf break, while it is the on-shelf winds that are responsible for the coastal downwelling.

“Holland et al. [28] show that greenhouse gas forcing has induced a transition from mean easterlies to near-zero mean winds in West Antarctica”. The context of this sentence within the whole paragraph implies to me that you're referring to the on-shelf winds that are responsible for coastal downwelling. However, Holland et al. are referring to the winds in a very specific location over the shelf break in the Amundsen Sea. In the same paper they show there is no significant GHG-driven trend in the easterly winds over the continental shelf. Therefore, I don't think this result from Holland et al. is quite relevant for explaining the potential lack of ASF in West Antarctica.

This is a good point, and we recognise we did not make enough distinction between on-shelf and shelf-break winds. We acknowledge that the time-mean winds over the shelf are indeed easterly, but they are still weak in comparison to the majority of East Antarctica where the ASF is generally more defined (see for example Caton-Harrison et al. 2022, Fig 1, Hazel et. al (2019), Fig 1).

We have rephrased the section in question. Firstly, we make the above point that coastal easterlies are relatively weak, which is a contributing factor in determining the strength of the ASF (from a time-mean perspective). Secondly, we acknowledge your point that the observable trend in these easterlies is not significant, and so have edited the GHG trend sentence to briefly state that future simulations indicate a region-wide weakening in easterlies (Neme et al. 2022) (Lines 75 - 77).

2. In much of the discussion of the undercurrent regime, I think the authors are implying that a necessary feature of the undercurrent regime is a weak or non-existent ASF, which I don't think is accurate. First, I would say that there is a reasonably strong ASF in parts of the Amundsen Sea (see the observations of Walker et al. (2013)), and that here the ASF is responsible for the existence of the undercurrent in the first place. Importantly though, I don't think whether or not an ASF exists in this region is of great importance for the authors' conclusions (I understand it's unclear to what extent the ASF exists across West Antarctica). The authors' important conclusion is that seasonal variability in the undercurrent comes from barotropic wind-driven forcing. Whether or not there is a time-mean ASF or whether or not it is seasonally varying is a separate from this.

I suggest that the authors ensure they don't state or imply that a lack of ASF is a necessary feature for seasonal variability in cross-slope transfer to be driven by undercurrent variability. In particular, I would remove or edit the sentence at L393 ("The ASF is not..."), as this could imply that a lack of ASF is important for the undercurrent regime to exist. I make the same comment for the sentence at L528 ("a lack of strong...").

Thank you, we agree that it is a mistake to imply that the existence of undercurrent-driven variability is attributable to a lack of ASF. We have removed L393 and L528 as suggested.

3. L320. Be careful with conflating winds and wind stresses. They are not necessarily oriented in the same direction, because of the sea ice. Can you clarify if here you are discussing the stress going into the ocean, through the combined effect of winds and sea ice? In Fig. 4 the title says "zonal wind", but caption says "wind stress". Which one is it? As I suggest above, if it's a stress, clarify as well if this is the actual stress going into the ocean, or some wind-only (ignoring sea ice) stress. I would be cautious of drawing any conclusions if the latter is used, as sea ice is a critical factor in ocean surface stresses.

It is 10m zonal wind velocity, not a stress. We have removed mention of stress in the relevant captions as this was mistakenly added.

Minor

Title: "Seasonal Regimes of Warm Circumpolar Deep Water Intrusion onto Antarctic Ice Shelves". I wouldn't describe CDW as intruding "onto" ice shelves. Perhaps "towards".

Done.

I don't think abstract should have a section number. E.g., the introduction is section 1.

Done.

L52. 'ice shelf'  'ice shelves'

Done.

L54. 'continent'  'continental'

Done.

L98. “although it is also possible that, at longer timescales, buoyancy forcing changes may be dominant in warm regimes [32][33].” I’m not sure that Silvano et al. (2022, your ref. [32]) are talking about changes in buoyancy forcing in their paper, but rather baroclinic adjustments to variability in the wind stress curl (mechanical forcing). However, Haigh & Holland (2024) do show buoyancy forcing to be the driver of variability on longer (decadal) timescales in a warm shelf regime.

Good point, thank you. We have updated the reference accordingly.

Fig 1. Briefly mention the depth of the dotted pink line. Is it possible to add shelf break contour to the temperature maps at the level of the pink line?

Done.

L163. Suggested changes: ‘The model employs a Mercator projection...’. And ‘The model has 225 vertical levels.’

Done.

L406. ‘negative zonal flow anomalies’. State which direction (westward) rather than negative.

Done.

Fig 7. Does the correlation in Fig. 7c get stronger when the bottom flow speed is lagged behind the undercurrent speed? What are the advective timescales involved?

We contend that the lag is likely to be sub-monthly. The along-canyon flows are part of the undercurrent system, which accelerates quasi-synchronously as it is a type of barotropic adjustment along f/H contours. We add such a statement in lines 266-271.

L433. “the Adelie shelf is sufficiently narrow to sustain a seasonal cycle of DSW formation and export”. Could you add just a few extra words to this sentence? Is your reasoning that the Adelie shelf fills with DSW relatively quickly, maybe over a few months, so DSW export has to occur on these seasonal timescales?

Yes, essentially this is our reasoning. This is explained in slightly more detail earlier in the manuscript (lines 173-177) and in far greater detail in Schmidt et al. (2023), which we reference. The authors show that, in regions where the shelf is very wide (Weddell, Ross), a multi-year reservoir of DSW can build up which minimises substantial seasonality in DSW concentrations there. Conversely, they show that, in the narrower shelves of Prydz Bay and the Adelie Coast, DSW reservoirs cannot persist longer than ~1 year. This is consistent with our seasonal variability metric in Figure 3, which shows substantial DSW variability on the narrow shelves, but almost none on the wider shelves. For clarity, we now reference Schmidt et al. in this line (originally L433, now 278 as methods moved to end).

Fig 10. It isn't clear in the upper panels: is the area shown wholly on the continental shelf?

Yes, and we now state this in the caption.

Response to Reviewer 3

I am concerned about the statistical analyses. The authors are limited by the model to a 1-year period (Line 168) which they analyze using monthly averages (Line 259). Based on e.g. Fig.8(c),9(c),10(c),11(c), the timeseries of model results are often dominated by a seasonal cycle. Quoting Emery & Thomson 2004:

"Since many data collected in time or space are highly correlated because of the presence of low-frequency, nearly deterministic components such as [...] the seasonal cycle, standard statistical methods do not really apply. Contrary to the requirements of stochastic theory, the values are not statistically independent. [...] A good example of this problem is presented by Chelton (1982) who showed that the high correlation between the integrated transport through Drake Passage in the Southern Ocean and the circumpolar-averaged zonal wind stress may largely be due to the presence of a strong semi-annual signal in both time series. A strong statistical correlation does not necessarily mean there is a cause and effect relationship between the variables."

So there is this first aspect that the correlations reported in the study may not be the causality that the authors are suggesting.

We agree that correlation does not necessarily imply causation. We have now explicitly acknowledged this in the text when we first introduce the correlation analysis (Lines 162-163).

A second aspect has to do with statistical significance and the concept of degrees of freedom, which I believe the authors are misusing. For their 1-year long monthly timeseries (N=12), the authors "assume 10 degrees of freedom" (caption of Figs.4,6,7,8,9,10,11). But as stated above, a correlation requires the values within the timeseries to be statistically independent, which will not be the case for a sinusoid function of a 1 year period that has been sampled every month (i.e. a timeseries dominated by the seasonal cycle; Fig.8(c),9(c),10(c),11(c)). Emery & Thomson 2004 suggest in this case to compute the *effective* number of degrees of freedom that takes into account the presence of correlation between neighboring monthly values. The $N-2=10$ degrees of freedom assumed in the study is an overly optimistic estimate that assumes all points to be statistically independent.

Thank you. In most cases, we retain significance even if we substantially reduce the number of degrees of freedom.

For the DSW-CDW correlations: In the captions of Figures 8,10, and 11, we now state the minimum number of degrees of freedom which retain significance at the 99% level. This is 3 dfs in Figure 8 and 7 dfs in Figures 10 and 11. Note that if we were to choose a lower significance threshold at 95%, then this number drops to 2 and 5 respectively (also stated in the figure captions).

For the CDW-wind correlations in Figure 4: we have added additional panels to Figure S4 to show the significance of the correlations assuming 10, 8, 6, and 4 degrees of freedom. Most regions which are significant at 10 degrees of freedom in Figure 4 retain significance until at least 6 degrees. We now state this in the caption of Figure 4, and reference the new panels of Figure S4 here as well.